# Preliminary evaluation of computer-assisted home training for French cochlear implant recipients

**Sandrine Kerneis**[1]*, **John J. Galvin, III**[1,2], **Stephanie Borel**[1,3], **Jean Baqué**[1], **Qian-Jie Fu**[4], **David Bakhos**[1,2,5]

1 University Hospital Center of Tours, FRA, Tours, France, 2 House Institute Foundation, Los Angeles, California, United States of America, 3 Assistance Publique Hôpitaux de Paris, Pitié-Salpêtrière and Sorbonne University, FRA, Tours, France, 4 Department of Head and Neck Surgery, David Geffen School of Medicine, UCLA, Los Angeles, California, United States of America, 5 INSERM UMR 1253 I-Brain, Université François-Rabelais de Tours, CHRU de Tours, FRA, Tours, France

* sandrine.kerneis@gmail.com

**Data Availability Statement:** All relevant data are within the paper and its Supporting information files.

## Abstract

For French cochlear implant (CI) recipients, in-person clinical auditory rehabilitation is typically provided during the first few years post-implantation. However, this is often inconvenient, it requires substantial time resources and can be problematic when appointments are unavailable. In response, we developed a computer-based home training software ("French AngelSound™") for French CI recipients. We recently conducted a pilot study to evaluate the newly developed French AngelSound™ in 15 CI recipients (5 unilateral, 5 bilateral, 5 bimodal). Outcome measures included phoneme recognition in quiet and sentence recognition in noise. Unilateral CI users were tested with the CI alone. Bilateral CI users were tested with each CI ear alone to determine the poorer ear to be trained, as well as with both ears (binaural performance). Bimodal CI users were tested with the CI ear alone, and with the contralateral hearing aid (binaural performance). Participants trained at home over a one-month period (10 hours total). Phonemic contrast training was used; the level of difficulty ranged from phoneme discrimination in quiet to phoneme identification in multi-talker babble. Unilateral and bimodal CI users trained with the CI alone; bilateral CI users trained with the poorer ear alone. Outcomes were measured before training (pre-training), immediately after training was completed (post-training), and one month after training was stopped (follow-up). For all participants, post-training CI-only vowel and consonant recognition scores significantly improved after phoneme training with the CI ear alone. For bilateral and bimodal CI users, binaural vowel and consonant recognition scores also significantly improved after training with a single CI ear. Follow-up measures showed that training benefits were largely retained. These preliminary data suggest that the phonemic contrast training in French AngelSound™ may significantly benefit French CI recipients and may complement clinical auditory rehabilitation, especially when in-person visits are not possible.

**Funding:** The authors received no specific funding for this work.

**Competing interests:** The authors have declared that no competing interests exist.

## Introduction

After cochlear implantation, post-lingual cochlear implant (CI) recipients must adapt to the new electrical speech patterns relative to central templates developed during previous acoustic hearing. Usually, the greatest adaptation occurs during the first 3–6 months of implant use, beyond which there is little further improvement [1–4]. Despite this initial adaptation, there is considerable variability in CI outcomes. Some CI users receive little benefit from the latest technology, and many have difficulty with challenging listening environments.

While passive learning (i.e., no explicit training) may drive adaptation during the first 3–6 months of CI use, computer-based auditory training (i.e., explicit feedback) has been shown to improve speech understanding, even in CI users with years of device experience [5–15]. Active training may better engage CI users' auditory plasticity and facilitate and/or accelerate adaptation. However, the method and stimuli used for auditory training also matter. Training with sentences (in quiet or in noise) largely targets top-down processes, where CI users may rely strongly on contextual cues for speech understanding. Compared to phoneme recognition, sentence recognition has been shown to be more robust to the spectro-temporal distortion associated with CIs [16]. Training at the phoneme level may target more bottom-up processes, where feedback can help CI recipients adapt to the spectro-temporal distortion (especially the acoustic-to-electric frequency mismatch) with electric hearing.

Fu and colleagues developed a computer-based research platform to provide training to English-speaking CI users in the lab or at home, using their own computers [5–8]. More than 4500 monosyllable words spoken by multiple talkers were recorded for the training software. Moderate amounts of monosyllable word training (~10 hours) that targeted phonemic contrasts significantly improved CI users' vowel and consonant recognition by ~15 percentage points, with significant improvements also observed for sentence recognition in noise [5–10, 14]. Early research versions of the software developed by Fu and colleagues were modified to create "AngelSound™" training software, which has been freely available for many years to CI recipients, clinicians and researchers. Worldwide, the software has been requested and/or downloaded by more than 100,000 CI recipients, clinicians and researchers. AngelSound™ is currently available in English and Mandarin language versions.

In France, post-implantation auditory rehabilitation programs are available during the first few years of CI use. These may not be ideal in that they require in-person visits with local speech therapists; during the recent COVID-19 pandemic, such non-essential in-person interactions were greatly curtailed. Methods are also not consistent across CI centers. In general, there is no theoretically based approach, and no opportunity to train within challenging listening environments such as noise or competing speech. Computer-based auditory training (as in AngelSound™) allows CI users to train at home at their convenience, with opportunities to adjust the level of difficulty according to individual needs (e.g., training in quiet or in noise). As noted above, moderate amounts of computer-based home training have been shown to significantly improve CI users' vowel, consonant and sentence recognition, with an average improvement of ~15 percentage points [8]. Historically, such post-training improvements are often as large if not larger than those brought about by improvements in CI technology. While previous studies have demonstrated the efficacy of computer-based home training [5–12, 14, 15], no such training is currently available for French CI recipients. To address this issue, we recently developed a French version of AngelSound™ that was similar in structure to previous English and Mandarin language versions. Stimuli consist of high-quality recordings (44.1 kHz sampling rate, 16-bit resolution) of French phonemes, words and sentences produced by multiple native French talkers.

While computer-based home training has been shown to benefit unilateral CI users, auditory rehabilitation strategies may be different for bilateral (CI in both ears) and bimodal listeners (CI in one ear, hearing aid in the other ear). For bilateral CI users (especially for sequentially implanted), speech understanding may be better in one ear than in the other. This performance asymmetry has been shown to limit the binaural benefit in bilateral CI users when speech and noise are presented directly in front of the listener [17, 18]. It is possible that the peripheral representations may differ due to different electrode positions and or patterns of neural survival in each ear, and the ear with the better representation may dominate the binaural perception [19]. When speech performance was similar across ears, the binaural benefit (i.e., gains in binaural performance relative to the better ear) was significantly greater [17, 18]. For sequentially implanted CI recipients, speech patterns developed with the first CI may dominate the binaural perception. Given that performance asymmetry may limit the binaural benefit, training the poorer ear may improve performance, reduce asymmetry, and increase binaural benefit.

Increasing numbers of CI recipients have residual acoustic hearing in the non-implanted ear. Depending on the extent of acoustic hearing in the non-implanted ear and the position and frequency allocation in the CI ear, there may be an inter-aural frequency mismatch. While unilateral CI users may adapt to an intra-aural frequency mismatch over time, bimodal CI users may have difficulty adapting to the mismatch in the CI ear, given the dominance of the acoustic hearing ear [20]. When hearing is limited in the acoustic ear, there is often large performance asymmetry due to loss of high-frequency speech information. Performance asymmetry has been shown to significantly limit binaural benefit in bimodal CI users [21]. Reducing the inter-aural frequency mismatch (which likely leads to performance asymmetry) has been shown to significantly improve binaural benefit in simulations of bimodal listening [22]. Training the CI ear alone may improve adaptation to the frequency mismatch, thereby improving integration of acoustic and electric hearing and thus, binaural benefit.

In the present study, we evaluated the efficacy of computer-based home training with French AngelSound™ in adult, native French-speaking unilateral, bilateral and bimodal CI recipients. Outcome measures included vowel recognition in quiet, consonant recognition in quiet, and sentence recognition in noise, as well as the shortened Speech, Sound and Quality questionnaire (SSQ-12; [23]). Unilateral CI users were tested with the CI alone. Bilateral CI users were tested with each CI ear alone to determine the poorer ear to be trained and with both ears (binaural performance). Bimodal CI users were tested with the CI ear alone and with the contralateral hearing aid (binaural performance). Participants were trained at home over a one-month period (10 hours total training time). Unilateral and bimodal CI users trained with the CI alone; bilateral CI users trained with the poorer CI ear alone. Outcomes were measured before training was begun, immediately after training was completed, and one month after training was stopped. Consistent with previous studies, we expected that training with the CI alone would improve performance with the CI alone. We were interested in whether training a single CI would also improve binaural performance in bilateral and bimodal CI users and, if so, whether binaural improvements would be super-additive (greater than CI-only improvements) or simply driven by CI-only improvements.

## Methods

### Ethics statement

All participants provided written informed consent prior to enrolling in the study. The study was approved by the Ethics committee at the University of Tours (Approval number: 2019–067).

## Participants

Fifteen adult CI users (7 male, 8 female; mean age at testing = 51.1 ± 15.3 years) were recruited from the ENT center at the University Hospital Center of Tours, France between June 2020 and February 2021. Inclusion criteria were: 1) Native French speaking, 2) More than 6 months of experience with their CI, 3) Ability to use a computer, follow instructions, and train at home. Table 1 shows demographic information. Five participants were unilateral CI users, 5 were bilateral CI users, and 5 were bimodal CI users. All participants' CI settings were checked by an audiologist before enrolling in the study, and participants were instructed not to change these settings during the study. The audiologist explained the study to the participants, who were then referred to one of two researchers who verified inclusion criteria, obtained written informed consent, and planned the three upcoming study appointments.

Sixteen adults with normal hearing (NH; mean age at testing = 42.5 ± 17.9 years) were recruited as experimental controls for the CI test data; vowel and consonant recognition were measured in 16 listeners with NH, and sentence recognition in noise was measured in 13 listeners with NH. A one-way ANOVA showed no significant difference in age at testing between CI and NH participants [$F_{(1,29)}$ = 2.0; p = 0.164].

**Table 1. Demographic information for the 15 CI participants.**

| Participant | Age test (yrs) | Sex | Hearing | Etiology | Dur deaf (yrs) | CI exp (yrs) | CI device | PTA | SAT | WRS |
|---|---|---|---|---|---|---|---|---|---|---|
| UNI-1 | 69 | M | Post | SHL—L Presbycusis—R | 10 | 5 | Coch | 28 | 38 | 100 |
| UNI-2 | 46 | F | Post | SHL—L Neurinoma—R | 1 | 5 | Coch | 18 | 25 | 100 |
| UNI-3 | 51 | F | Post | Unknown | 15 | 2 | Coch | 15 | 30 | 100 |
| UNI-4 | 55 | M | Post | Unknown | 29 | 6 | Coch | 27 | 30 | 100 |
| UNI-5 | 30 | F | Post | Unknown | 19 | 6 | Med-El | 35 | 47 | 60 |
| BI-1 | 55 | M | Post | Menières | L—18 | L—2 | Coch | 32 | 30 | 100 |
| | | | | | R– 28 | R—12 | | | | |
| BI-2 | 59 | M | Pre | Unknown | L—55 | L—4 | Coch | 27 | 32 | 100 |
| | | | | | R—57 | R—2 | | | | |
| BI-3 | 18 | F | Pre | Congenital | L- 14 | L—4 | AB | 33 | 43 | 90 |
| | | | | | R -16 | R—2 | | | | |
| BI-4 | 63 | F | Post | Ototoxicity | L—20 | L -10 | Coch | 18 | 28 | 100 |
| | | | | | R—23 | R -7 | | | | |
| BI-5 | 26 | M | Post | Cogan syndrome | L—0.5 | L—1 | Coch | 27 | 27 | 100 |
| | | | | | R—0.5 | R—1 | | | | |
| BM-1 | 69 | F | Post | Presbycusis—L SHL—R | 5 | 3 | Coch | 27 | 27 | 100 |
| BM-2 | 60 | M | Post | Chronic otitis | 41 | 2 | AB | 27 | 52 | 90 |
| BM-3 | 56 | F | Post | Hereditary | 21 | 5 | Coch | 33 | 33 | 100 |
| BM-4 | 49 | F | Post | Unknown | 18 | 2 | AB | 30 | 36 | 100 |
| BM-5 | 60 | M | Post | Hereditary | 6 | 2 | Coch | 13 | 24 | 100 |

CI = cochlear implant; UNI = unilateral CI; BI = bilateral CI; BM = bimodal CI; yrs = years; Post = post-lingual deafness; Pre = pre-lingual deafness; SNHL = sensorineural hearing loss; dur deaf = duration of deafness before CI; CI exp = CI experience; Coch = Cochlear© device; AB = Advanced Bionics© device; PTA = aided (CI) pure-tone average threshold (across 500–1000–2000 kHz) in dB HL (hearing level); SAT = speech audibility threshold in dB HL; WRS = percent correct monosyllable word recognition score in quiet

## Outcome measures

Outcome measures included phoneme recognition in quiet and sentence recognition in steady noise, as well as the SSQ-12 questionnaire [23]. Baseline measures (vowel and consonant recognition in quiet, sentence recognition in noise) were measured before training was begun ("pre-training"). Participants then began the home training for 1 month (10 hours total training), immediately after which outcomes were measured again ("post-training"). After completing post-training measures, participants were asked to discontinue any training with French Angel-Sound™. One month after training was stopped, outcomes were re-measured to observe whether any training benefits had been retained ("follow-up"). Fig 1 illustrates the timeline of the study.

Due to the generally high variability in CI speech performance, it can be difficult to separate within-subject training effects from across-subject variability. To avoid this, a within-subject control procedure was adopted instead of a separate control group.

All pre-training, post-training and follow-up tests were performed in a soundproof booth that was regularly used for evaluations and that CI participants were familiar with. Testing was conducted by a researcher who was unknown to participants before the study started. Only the participant and the researcher were present in the room.

Vowel and consonant test stimuli were digitized natural productions (44.1 kHz sampling rate, 16-bit resolution) from two adult, native French-speaking talkers (one man and one woman) that were not included in the training stimuli. Vowels included 10 tokens presented without context (vowel only), for a total of 20 stimuli in the stimulus set. Consonants included 19 tokens presented in /a/-consonant-/a/ context, for a total of 38 stimuli in the stimulus set. Vowel recognition in quiet was measured using a 10-alternative, forced-choice (10AFC) closed-set paradigm. Consonant recognition in quiet was measured using a 19AFC closed-set paradigm. Vowel and consonant stimuli were presented over a single loudspeaker at 65 dBA, with the listener seated facing the loudspeaker 1 m away. During testing, a stimulus was randomly selected from the stimulus set (without replacement) and presented to the listener, who responded by clicking on one of the response choices shown on a computer screen (see Fig 2 for the vowel and consonant response screens). After selecting the response, the next token was presented. For each test run, the stimulus set was presented once. No feedback was provided, and listeners were instructed to guess if they were unsure of the correct answer. For each participant, 3–4 test runs were performed for vowel recognition and 2–3 test runs were performed for consonant recognition. Data were averaged across all test runs.

Speech recognition thresholds (SRTs), defined as the signal-to-noise ratio (SNR) that produces 50% correct sentence recognition in steady noise [24], were measured using French

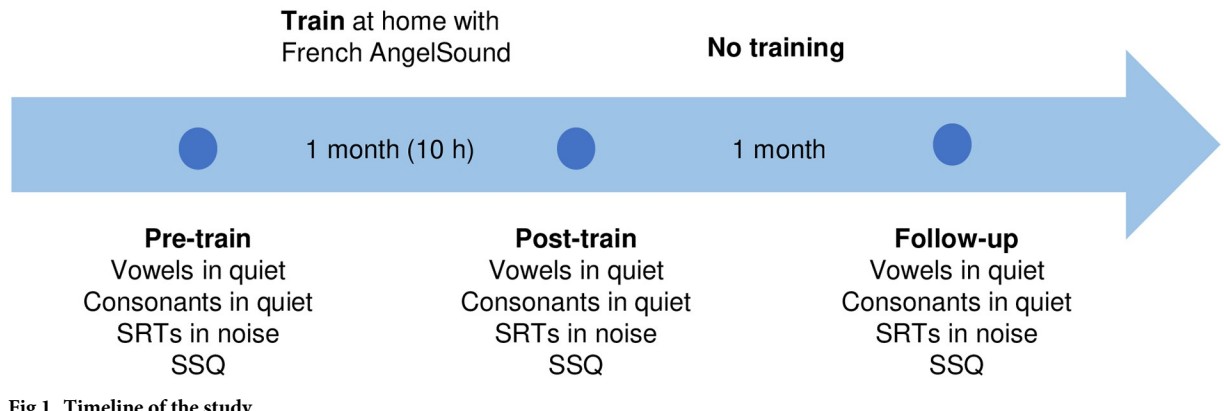

**Fig 1. Timeline of the study.**

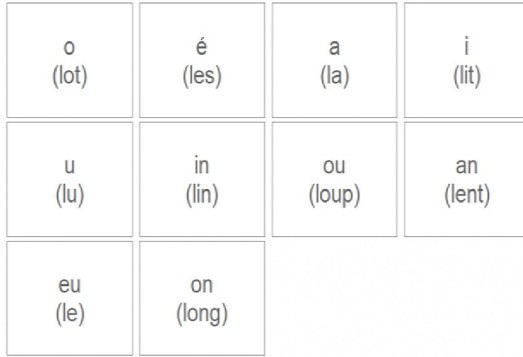 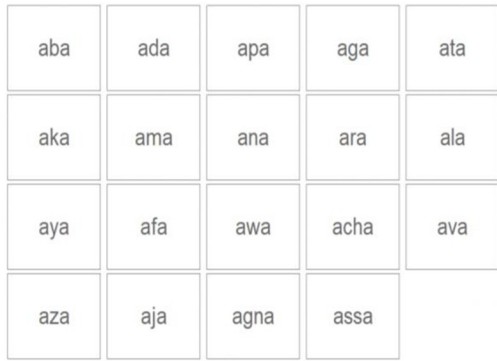

**Fig 2. Response screens showing the stimuli used for the vowel (left) and consonant recognition tests (right) in quiet.**

Hearing-in-noise-test (HINT) sentences [25]. The sentence stimuli used for this study were new recordings produced by the same male and female talkers that produced the vowel and consonant test stimuli (44.1 kHz sampling rate, 16-bit resolution). White noise was filtered to match the combined long-term spectrum across all test sentences. SRTs were adaptively measured using an open-set paradigm. Sentences and noise were presented from a single loudspeaker; the listener was seated facing the loudspeaker 1 m away. The onset and offset of the noise were 500 ms before and after the sentence, respectively. The level of the sentence was fixed at 65 dBA, and the noise level was adapted. The total stimulus set consisted of 100 sentences. For each test run, 20 sentences were randomly selected from the total set. During testing, a sentence was randomly selected from the 20-sentence set (without replacement) and presented at the designated SNR. The listener responded by repeating as many words as heard, and the experimenter marked the correct responses. The SNR was adjusted according to the correctness of response using "Rule 3" from Chan et al. [26]: if 50% or more of the words were repeated correctly, the SNR was reduced; if not, the SNR was increased. If the SNR exceeded 25 dB during a test run, the run was aborted. The initial SNR was 10 dB. The initial step size was 4 dB for the first two reversals in SNR, and the final step size was 2 dB. The first sentence was played until the listener correctly repeated 50% of the words. If the listener failed to repeat 50% of the words, the sentence was played again and the SNR was increased (in 4 dB steps) until 50% of the words were repeated correctly. If the SNR exceeded 25 dB, the test run was aborted and a new test run was begun with a new initial sentence. After correctly repeating 50% of the words in the first sentence, the remaining 19 sentences in the test run were presented one time. The final six reversals in SNR were averaged to calculate the SRT for the test run. Three test runs were performed for each listening condition (CI-only for unilateral and bimodal CI users; binaural for bilateral and bimodal CI users).

For unilateral CI users, vowel recognition was measured first, followed by consonant recognition and SRTs. For bilateral CI users, binaural vowel recognition was measured first to familiarize participants with the task using their everyday listening condition. Next, vowel recognition was measured in each CI ear alone to determine the poorer ear that would be used for training. Next, binaural consonant recognition and SRTs were measured; due to time constraints, consonant recognition and SRTs were measured only with binaural listening. For bimodal CI users, binaural vowel recognition was measured first to familiarize participants with the task using their everyday listening condition, followed by CI-only vowel recognition. Next, binaural consonant recognition was measured, followed by CI-only consonant

recognition. Next, binaural SRTs were measured, followed by CI-only SRTs. For CI-only performance, the contralateral hearing aid was removed, and the ear was plugged.

## Training protocol

Training was performed using the recently developed French version of AngelSound™, which is similar in structure to the previous English and Mandarin versions of AngelSound™. For the phonemic contrast training used in the present study, there were more than 1000 French monosyllable words produced by 2 male and 2 female native French talkers; these training stimuli were not used for testing. All vowel and consonant stimuli were tagged with relevant acoustic features that allowed for adjustment to the degree of training difficulty to include relatively strong or weak acoustic contrasts. For the vowel stimuli, acoustic features included first formant (high, middle, or low), second formant (front or back), third formant (rounded or unrounded), and nasality (absent or present). For the consonant stimuli, acoustic features included voicing (voiced or unvoiced), manner (stop, nasal, fricative, lateral approximate, or semi-vowel), and place of articulation (front, middle, or back).

After completing baseline measures, participants were provided with a study-specific version of French AngelSound™ via web link or on a USB thumb drive; the software allowed for only phonemic contrast training, and all other training modules were disabled. Participants were extensively instructed how to install the software and how to perform the training. None reported any difficulty installing or using the training software.

The training protocol was similar to that used in previous studies by Fu and colleagues [5–8, 11, 12, 15, 27]. Participants were asked to train for 30 minutes per day, 5 days per week for 4 weeks, resulting in 10 hours of total training. The training software logged all time spent training. After completing the home training, participants returned to the lab and vowel recognition, consonant recognition, and SRTs were re-measured; participants also completed the SSQ-12 questionnaire. After completing these post-training outcome measures, training was stopped. One month after training was stopped, participants returned to the lab for follow-up measures to observe whether any post-training benefit had been retained.

Participants were asked to train with the vowel and consonant stimuli. Participants were instructed to train in a quiet environment with the computer speaker output set to a comfortably loud level. Unilateral and bimodal CI users were instructed to train with the CI ear only; bilateral CI users were instructed to train with the poorer ear only (identified from baseline testing results), or with the more recently implanted ear if performance was similar across ears. Participants were also instructed that they could break up each day's training into multiple session (e.g., 3 sessions of 10 minutes each) to accommodate their schedule, attentiveness, fatigue, etc. Each training run consisted of 25 stimuli and took 5–10 minutes to complete. Results were recorded only when a training run was completed. After completing the home training, training data were shared with the experimenter, including the total time spent training, the levels of difficulty trained, and the percent correct for each training run.

Pre-training vowel and consonant scores were used to suggest the appropriate level of difficulty. This was recommended as the most appropriate training level, but participants were free to train on any level of difficulty that they desired. According to participant interviews, the majority of participants trained with all levels, typically using the recommended levels in the beginning, and increasing the difficulty when they wanted to diversify their training and decreasing it if they found it too difficult. Within each level of difficulty, there were multiple sublevels of difficulty in which the phonemic contrasts were reduced. Levels of difficulty included:

• Level 1 (Easy): Simple discrimination of phonemes in words. In each trial, three sounds were played in sequence; two of the sounds were identical and the third was different. The three

sounds were randomly assigned to three intervals (labeled "Son 1," "Son 2", Son 3"), and participants were asked to choose which sound was different. Visual feedback was provided as to the correctness of response. If the participant did not respond correctly, auditory and visual feedback were provided in which the three sounds, now labeled with the appropriate words (e.g., "gare" [gaʁ], "guerre" [gɛʁ], "toque" [tɔk], and "tek" [tɛk] were replayed in the same sequence, allowing the participant to compare their response to the correct response. Within Level 1, sublevels of difficulty were increased by reducing the phonemic contrast across response options.

- Level 2 (Easy to Medium): Identification of phonemes in words with audio-visual preview. In each trial, two stimuli were shown onscreen (with the appropriate word labels) and played in sequence. One of the two stimuli was then played using a different talker than for the preview, and participants clicked on the correct word. Only one phoneme differed between response choices (e.g., "toque", "tek"). Visual feedback was provided as to the correctness of response. If the participant did not respond correctly, auditory and visual feedback were provided in which the two stimuli, produced by the different talker and labeled with the appropriate words (e.g., "toque", "tek"), were replayed, allowing the participant to compare their response to the correct response. Within Level 2, sublevels of difficulty were increased by reducing the phonemic contrast across response options.

- Level 3 (Medium): Identification of phonemes in words without preview. In each trial, two response choices were shown onscreen, with only one phoneme differing between response choices (e.g., "toque", "tek"). A stimulus was played that corresponds to one of the response choices, and participants clicked on the response choice that best matched the stimulus. Visual feedback was provided as to the correctness of response. If the participant did not respond correctly, auditory and visual feedback were provided in which stimuli corresponding to the two response choices were played back, allowing the participant to compare their response to the correct response. Within Level 3, sublevels of difficulty were increased by reducing the phonemic contrast across response choices and by increasing the number of response choices.

- Level 4 (Medium to Difficult): Identification of phonemes in words without preview. Level 4 was similar to Level 3, except that the sublevels of difficulty were increased according to perceptual confusions (rather than acoustic contrasts) and by increasing the number of response choices.

- Level 5 (Difficult). Identification of phonemes in words in multi-talker babble without feedback. Level 5 was similar to Level 3, except that phoneme identification was measured in the presence of multi-talker speech babble. The number of response choices was fixed at 4. For each training run, identification was trained at a fixed SNR. A stimulus was played that corresponded to one of the response choices, and participants clicked on the response choice that best matched the stimulus. Visual feedback was provided as to the correctness of response. If the participant did not respond correctly, auditory and visual feedback were provided in which stimuli were played back at the fixed SNR, allowing the participant to compare their response to the correct response. Within Level 5, sublevels of difficulty were increased by reducing the SNR.

## Results

All raw data from the present study can be found in "S1 File".

### Phoneme recognition and SRTs in listeners with NH

Vowel and consonant recognition in quiet were measured in 16 listeners with NH, and SRTs in noise were measured in 13 listeners with NH. Mean vowel recognition was 99.4 ± 2.0 percent correct, mean consonant recognition was 99.5 ± 1.4 percent correct, and mean SRTs were -5.3 ± 2.2 dB.

### Compliance with the testing and training protocol

All participants returned to the lab for the required test appointments. Compliance with the training protocol was good, with a mean training of 31.7 ± 4.0 minutes per day (10.3 ± 1.7 total hours) during the one-month training period. Participants were instructed to immediately contact the experimenters if they had any problems installing or using the software, but no problems were reported. Most participants found the software easy to use and the training easy to perform, although some commented that it got a bit boring after a while. Nonetheless, 66% of participants spontaneously asked if they could keep the training software after the experiment was finished; these requests were honored.

### Pre-training, post-training, and follow-up measures in CI users

**Vowel recognition in quiet.** CI participants completed 3–4 test runs of vowel recognition at pre-training, post-training, and follow-up. Vowel recognition scores for individual participants as a function of test run are shown in "S1 Fig". Fig 3 shows mean pre-training, post-training, and follow-up vowel recognition scores (top) and training benefit (bottom) with CI-only (left) and binaural listening (right). Linear mixed model (LMM) analyses were performed on the CI-only vowel score data, with group (unilateral, bilateral, bimodal), test point (pre, post, follow-up), and test run (1, 2, 3, 4) as fixed factors and participant as a random factor. Results showed significant main effects for test point [$F_{(2, 98.5)}$ = 43.6, $p < 0.001$] and test run [$F_{(3, 98.2)}$ = 3.5, $p = 0.019$], but not for group [$F_{(2, 2.3)}$ = 3.5, $p = 0.141$]; there were no significant interactions. For unilateral, bilateral, and bimodal CI users, post-hoc Bonferroni pairwise comparisons showed that vowel scores were significantly higher at post-training and follow-up than at pre-training ($p < 0.05$ for both comparisons), with no significant difference between post-training and follow-up. For unilateral, bilateral, and bimodal CI users, vowel scores were significantly higher for test run 3 than run 1 ($p < 0.05$), with no significant difference between runs 2, 3, and 4.

LMM analysis was also performed on the binaural vowel score data, with group (bilateral, bimodal), test point (pre, post, follow-up), and test run (1, 2, 3, 4) as fixed factors and participant as a random factor. Results showed a significant main effect for test point [$F_{(2, 64)}$ = 53.2, $p < 0.001$], but not for test run [$F_{(3, 64.1)}$ = 1.2, $p = 0.306$] or group [$F_{(1, 8)}$ = 1.5, $p = 0.253$]; there were no significant interactions. Bonferroni pairwise comparisons showed that vowel scores were significantly higher at post-training and follow-up than at pre-training ($p < 0.05$ for both comparisons), with no significant difference between post-training and follow-up.

LMM analyses were also performed on the CI-only and binaural vowel training benefit data, with group (unilateral, bilateral, bimodal), test point (post, follow-up), and test run (1, 2, 3) as fixed factors and participant as a random factor. For both analyses, results showed no significant main effects.

**Consonant recognition in quiet.** CI participants completed 2–3 test runs of consonant recognition. Pre-training, post-training, and follow-up consonant recognition scores for individual participants as a function of test run are shown in "S2 Fig". Note that CI-only performance was not measured in bilateral CI users due to time constraints. Fig 4 shows mean pre-training, post-training, and follow-up consonant recognition scores (top) and training benefit

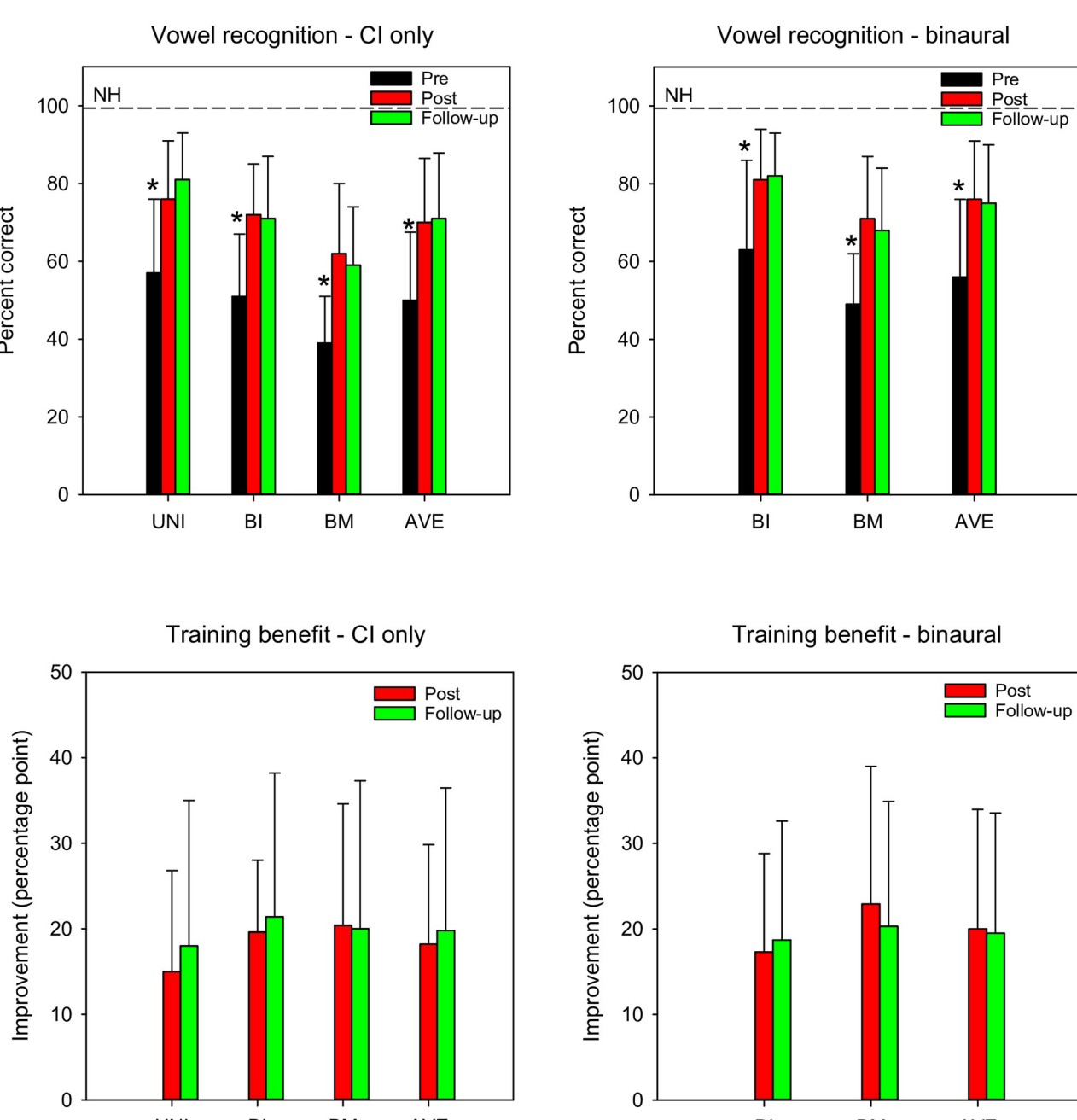

**Fig 3. Mean vowel recognition and training benefit.** The top panels show vowel recognition scores with CI-only (left) and binaural listening (right); the dashed lines show mean NH performance. The bottom panels show training benefit with CI-only (left) and binaural listening (right). The error bars show the standard deviation. The asterisks show significantly different scores.

(bottom) with CI-only (left) and binaural listening (right). LMM analyses were performed on CI-only consonant score data, with group (unilateral, bimodal) test point (pre, post, follow-up), and test run (1, 2, 3) as fixed factors and participant as a random factor. Results showed significant main effects for test point [$F_{(2, 56)} = 48.1$, $p < 0.001$], but not for test run [$F_{(2, 56)} = 0.7$, $p = 0.502$], or group [$F_{(1, 8)} = 4.8$, $p = 0.060$]; there were no significant interactions. For unilateral, bilateral, and bimodal CI users, post-hoc Bonferroni pairwise comparisons showed

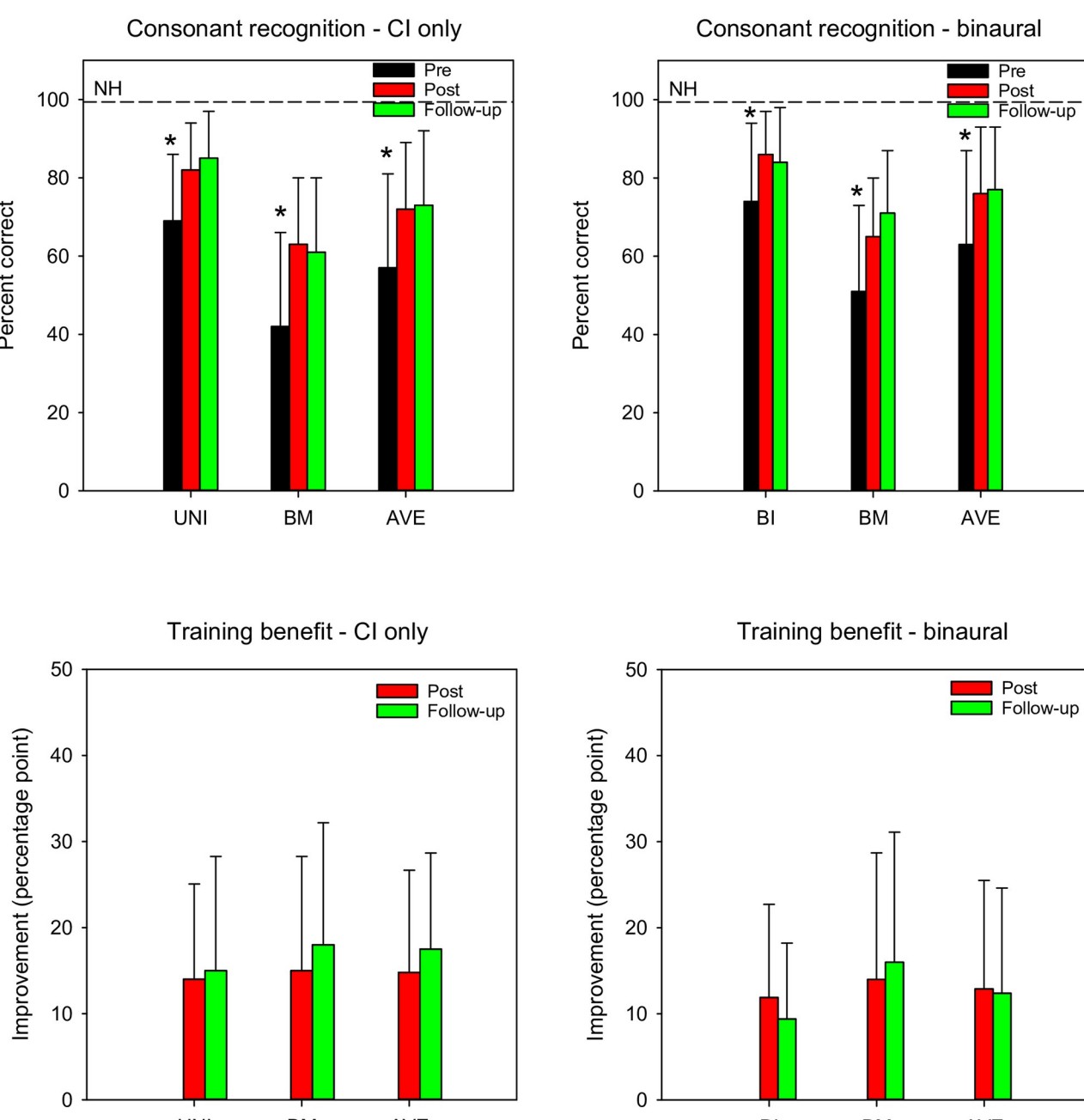

**Fig 4. Mean consonant recognition and training benefit.** The top panels show consonant recognition scores with CI-only (left) and binaural listening (right); the dashed lines show mean NH performance. The bottom panels show training benefit with CI-only (left) and binaural listening (right). The error bars show the standard deviation. The asterisks show significantly different scores.

that consonant scores were significantly higher at post-training and follow-up than at pre-training (p < 0.05 for both comparisons), with no significant difference between post-training and follow-up.

LMM analysis was also performed on the binaural consonant score data, with group (bilateral, bimodal), test point (pre, post, follow-up), and test run (1, 2, 3) as fixed factors and participant as a random factor. Results showed a significant main effect for test point [$F_{(2, 60)} = 29.8$,

$p < 0.001$], but not for test run [$F_{(2, 60)} = 2.3$, $p = 0.109$] or group [$F_{(1, 8)} = 3.1$, $p = 0.117$]; there were no significant interactions. Bonferroni pairwise comparisons showed that consonant scores were significantly higher at post-training and follow-up than at pre-training ($p < 0.05$ for both comparisons), with no significant difference between post-training and follow-up.

LMM analyses were also performed on the CI-only and binaural consonant training benefit data, with group (unilateral, bilateral, bimodal), test point (post, follow-up), and test run (1, 2, 3) as fixed factors and participant as a random factor. For both analyses, results showed no significant main effects.

**Sentence recognition in noise.** CI participants completed 3 test runs of sentence recognition in noise. Note that CI-only SRTs were not measured in bilateral CI users due to time constraints. Fig 5 shows SRT test run data from individual unilateral, bilateral, and bimodal CI users at pre-training, post-training, and follow-up. For pre-training measures, CI-only SRTs < 25 dB could not be obtained from UNI-1, UNI-2, BM-1, and BM-2. However, CI-only SRTs < 25 dB were obtained in these participants at post-train and follow-up. Pre-training binaural SRTs <25 dB could not be obtained in BM-1; again, binaural SRTs for this participant were < 25 dB at post-training and follow-up. The limited number of participants precludes any meaningful data analysis. In general, CI participants' SRTs were lower (better) at post-training and follow-up than at baseline. However, CI participants' SRTs remained much poorer than mean SRTs for listeners with NH (-5.3 ± 2.2 dB), even after training.

**Effect of training on binaural advantage.** While bilateral CI users were trained with the poorer CI-ear alone and bimodal CI users were trained with the CI-ear alone. This is not their everyday binaural listening experience, and binaural performance is generally better than CI-only performance. It was unclear whether the CI-only training would further increase the binaural advantage, or whether training benefits would similarly affect CI-only and binaural performance. For bilateral CI users, binaural advantage was calculated as the difference in vowel scores between binaural listening and CI listening with the poorer ear; note that only vowel recognition was measured with the CI ear alone due to time constraints. For bimodal CI users, binaural advantage was calculated as the difference in vowel and consonant scores between bimodal listening and CI listening. Fig 6 shows the mean binaural advantage for bilateral (vowels) and bimodal CI users (vowels and consonants) at pre-training, post-training, and follow-up. LMM analysis was performed on the binaural advantage data, with group (bilateral, bimodal), test (vowels, consonants), and test point (pre, post, follow-up) as fixed factors and participant as a random factor. Results showed no significant difference in bimodal advantage across groups, tests, or test points.

**Speech, spatial, and quality of hearing questionnaire.** Fig 7 shows mean pre-training, post-training, and follow-up SSQ-12 scores for the speech, spatial, and quality subsections, as well as the total SSQ-12 score. LMM analysis was performed on each of the subsections of the SSQ, with group (unilateral, bilateral, bimodal) and test point (pre, post, follow-up) as fixed factors and participant as the random factor. A significant effect of group was observed for the speech [$F_{(2, 36)} = 12.0$, $p < 0.001$], spatial [$F_{(2, 36)} = 6.2$, $p = 0.005$], and quality subsections [$F_{(2, 36)} = 6.0$, $p = 0.006$]. There was no significant effect test point and no significant interaction for any of the subsections. For all subsections, Bonferroni pairwise comparisons showed that SSQ scores were higher for the bilateral than for the unilateral and bimodal groups ($p < 0.05$ for both comparisons), with no significant difference between the unilateral and bimodal groups.

## Discussion

The computer-based home training showed significant benefits for vowel and consonant recognition for the present French CI participants, consistent with previous home-training

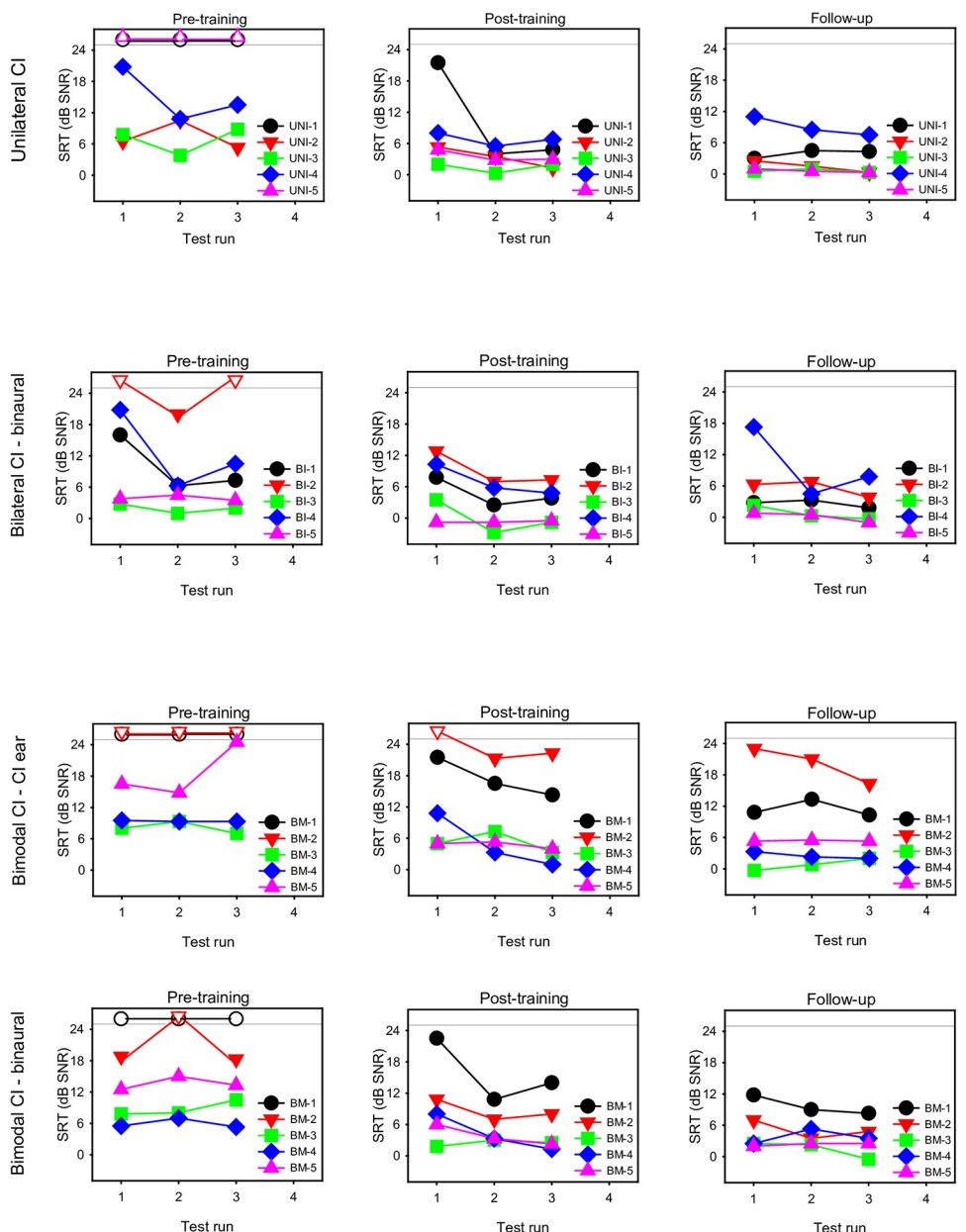

**Fig 5. SRTs for individual participants as a function of test run.** Pre-training (left column), post-training (middle column), and follow-up data (right column) are shown for the unilateral group (top row), bilateral group with binaural listening (middle row), and the bimodal group with CI-only and binaural listening (bottom two rows). The horizontal line shows the maximum SNR (25 dB) before the test run was aborted. The open symbols show test runs that exceeded the maximum SRT (25 dB SNR).

studies with English-, German- and Mandarin-speaking CI users [5–12, 14, 15]. For bilateral and bimodal CI users, training with a single CI ear improved both CI-only and binaural performance, consistent with Zhang et al. [12]. For all CI groups and listening conditions, follow-up measures were not significantly different from post-training measures, suggesting that training benefits had been largely retained, consistent with previous studies using a similar test and training protocol [11, 12, 28].

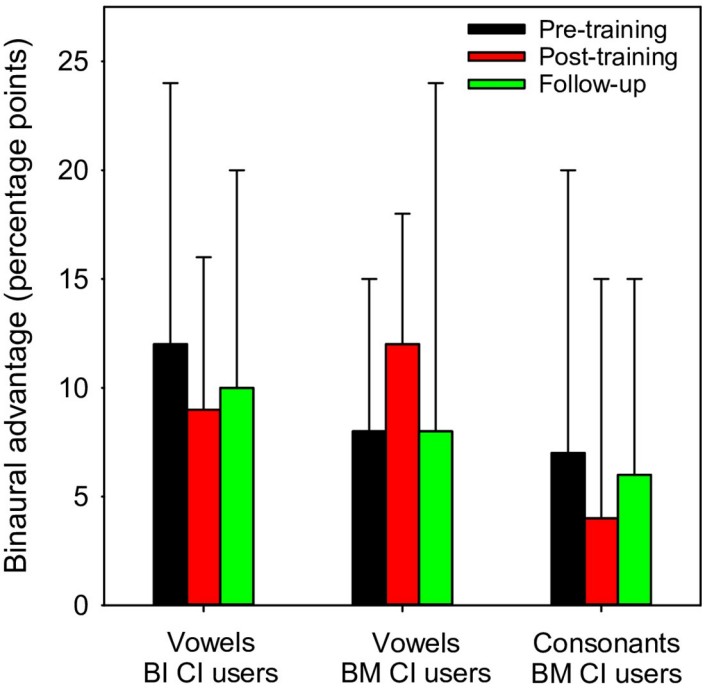

**Fig 6. Mean binaural advantage at pre-training, post-training, and follow-up.** Binaural advantage was calculated as the difference between binaural and CI-only performance. Data are shown for vowel recognition in bilateral CI users, and for vowel and consonant recognition in bimodal CI users. The error bars show the standard deviation.

### Benefits of training the CI ear for phoneme recognition

**CI-only listening.** Consistent with previous computer-based training studies for English-speaking [5–10, 27], German-speaking [14] and Mandarin-speaking CI users [28], the present study found significant benefits for phonemic contrast training in French CI users. With CI-only listening, the mean post-training benefit across all participants was 14.8 ± 11.9 and 14.8 ± 11.9 percentage points for vowel and consonant recognition, respectively. Post-training improvements with the present French participants were comparable to those reported in previous computer-based CI training studies using similar test and training protocols. For adult English-speaking unilateral CI users, Fu et al. [27] reported mean training benefits of 15.8 and 13.5 percentage points for vowel and consonant recognition, respectively. Wu et al. [28] reported mean training benefits of 21.7 (vowels) and 19.5 percentage points (consonants) in pediatric Mandarin-speaking CI (n = 7) and hearing aid (n = 3) listeners. Thus, the phonemic contrast training used in the present and previous studies [5–7, 27, 28] appears to produce consistent training benefits in terms of vowel and consonant recognition, even for experienced CI users.

**Binaural listening.** For bilateral and bimodal CI users, training with a single CI ear generalized to significantly improved binaural phoneme recognition. For bilateral CI users, the mean post-training improvement was 17.3 ± 11.5 and 11.9 ± 10.8 percentage points for binaural vowel and consonant recognition, respectively. For bimodal CI users, the mean post-training improvement was 22.9 ± 16.1 and 16.0 ± 15.1 percentage points for binaural vowel and consonant recognition, respectively. There was no significant difference in post-training improvements between CI-only and binaural listening, consistent with Zhang et al. [12].

Yoon et al. [17, 21] found that the greatest binaural benefit was observed when the performance asymmetry across ears was reduced. For the present bilateral CI users, mean CI-only

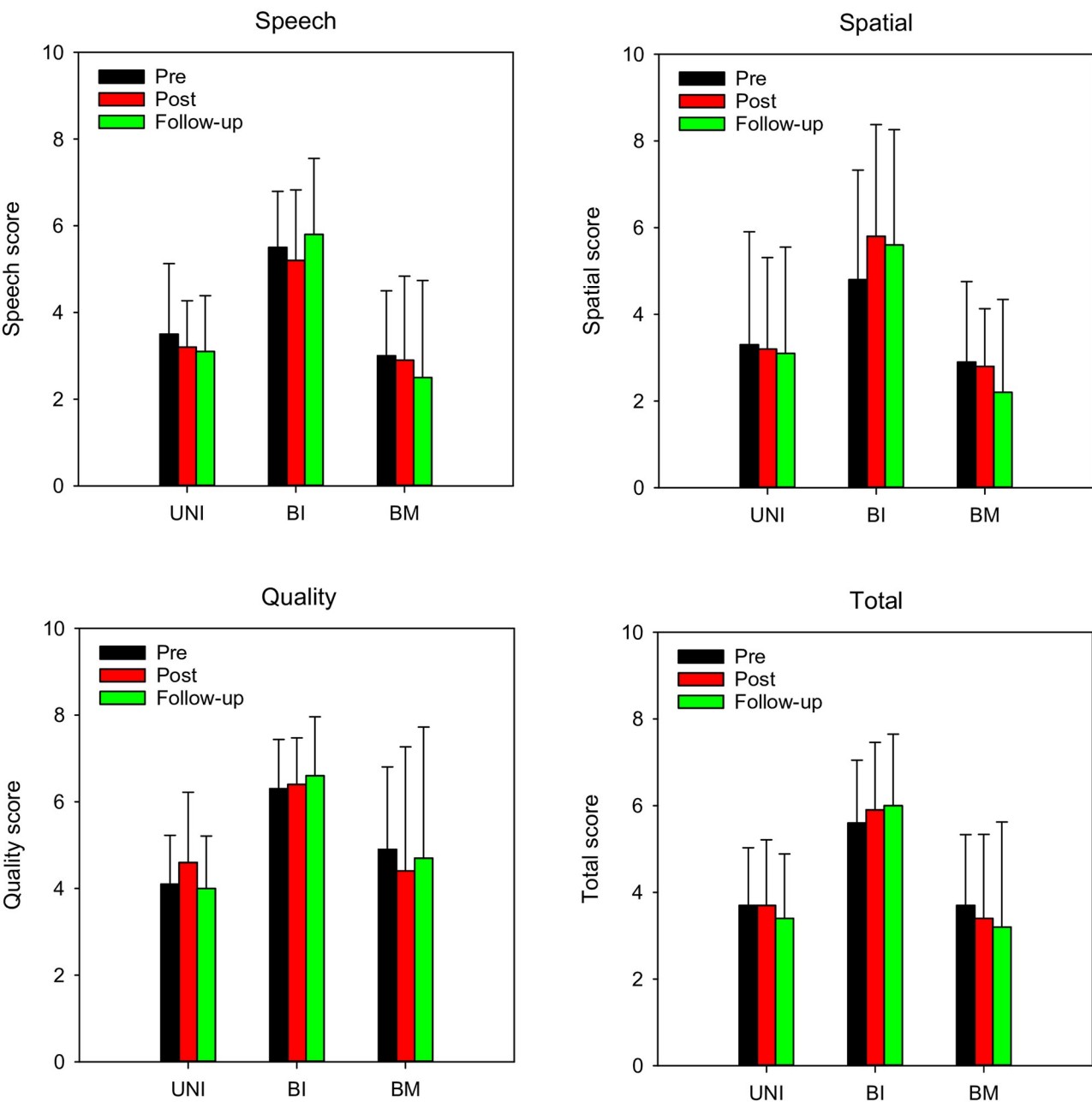

**Fig 7. Mean pre-training, post-training, and follow-up SSQ scores for the different CI groups.** Data are shown for the speech, (top left), spatial (top right), and quality subsections (bottom left), and for the total score (bottom right). In all panels, the error bars show the standard deviation.

vowel recognition was only 2.4 percentage points better with the better CI ear than with the poorer CI ear, with across-ear asymmetry ranging from 1.3 to 8.3 points. The mean post-training improvement for vowel recognition in bilateral CI users was quite similar between CI-only (19.6 ± 8.4 points) and binaural listening (17.3 ± 11.5 points). For bimodal CI users, performance with the non-implanted ear alone was not measured but would be expected to be poorer than with the CI ear, given the limited frequency extent. As such, substantial performance asymmetry across ears would be expected in bimodal CI users. Training with the CI ear

alone yielded similar post-training improvements for CI-only ($20.4 \pm 14.2$ and $15.0 \pm 13.3$ points for vowel and consonant recognition, respectively) and binaural listening ($22.9 \pm 16.1$ and $14.0 \pm 14.7$ points for vowel and consonant recognition, respectively).

Training the CI-ear alone produced similar improvements in CI-only and binaural performance. This suggests that post-training binaural improvements were likely driven by the improvements in the trained ear. The binaural advantage for bilateral and bimodal CI users was not significantly different across pre-training, post-training, and follow-up measures (Fig 6). As such, the performance asymmetry observed in bilateral and bimodal CI users before training largely remained after training despite the improvements in performance. Note that performance was not measured with the acoustic ear alone in bimodal CI users; we assumed that according to CI candidacy requirements, performance in the acoustic ear was sufficiently poor to support cochlear implantation, and remained poorer than that with the CI ear after implantation. It is unclear whether binaural training would have elicited similar or better improvements in CI-only and binaural listening. For example, Chavant et al. [29] found that for simulations of bimodal listening, including low-frequency acoustic information (even though unintelligible) during training benefited perception of spectro-temporally degraded speech.

## Benefits of training the CI ear only for SRTs in noise

**CI-only listening.** Pre-training CI-only SRTs < 25 dB could not be obtained in participants UNI-1, UNI-5, BM-1, and BM-2; pre-training binaural SRTs < 25 dB could not be obtained in BM-1. However, SRTs < 25 dB could be obtained in these participants at post-training and follow-up. These improvements in SRTs were sometimes suspiciously large (e.g., >20 dB improvement for UNI-5 at post-training and follow-up), and much greater than would be expected from the phonemic contrast training. In participants for whom CI-only pre-training SRTs <25 dB could be obtained, mean post-training improvements were $5.9 \pm 2.1$ and $7.0 \pm 5.9$ dB for unilateral and bimodal CI users, respectively.

Some issues with the sentence testing may have contributed to these patterns in results. First, there were only 100 French HINT sentences in the stimulus set. During each test trial, a sentence was randomly selected from the stimulus set, without replacement. Given the number of conditions (3 test times x 2 listening modes x 3 repeats), some sentences were inevitably repeated across test runs and/or test sessions. Given that listeners only had to correctly repeat 50% of the words to reduce the SNR (Rule 3 from Chan et al., 2008 [26]), previous exposure to sentences would have been beneficial. Second, the adaptive SRT procedure may have been problematic for some participants. Under the test protocol, the initial SNR was 10 dB. If the participant did not repeat 50% or more of the words from the first sentence correctly, the SNR was increased by the initial step size of 4 dB; this continued until reaching the maximum SNR of 25 dB (i.e., four opportunities to correctly identify 50% or more of the words in sentences before the test run was aborted). It is possible that this contributed to the failure to obtain pre-training SRTs <25 dB in some participants, and to aberrantly high SRTs in other participants. In previous studies by Fu and colleagues using English HINT sentences [11], such difficulties in obtaining SRTs in unilateral CI users using the same method were not observed. Third, it is possible that the present recordings of the French-Canadian HINT sentences [25] by native French speakers may have resulted in different intelligibility for some sentences. The mean SRT across both talkers ($-5.3 \pm 2.2$ dB) in the present native French listeners with NH was lower (better) than that reported by Vaillancourt et al. [25] in French-Canadian listeners with NH ($-3.0$ dB). Rule 3 was used in the present study, where listeners had to identify $\geq$50% of words in sentences to reduce the SNR [26]; Rule 1 was used in Vaillancourt et al. [25], where listeners had to identify 100% of words in sentences to reduce the SNR.

**Binaural listening.** Pre-training binaural SRTs <25 dB could be measured in 4 of 5 bimodal CI users and in all 5 bilateral CI users. For both groups, binaural SRTs were generally lower at post-training and follow-up than at pre-training. For the three bimodal CI users where CI-only SRTs were <25 dB (BM-3, BM-4, BM5), the mean post-training improvements were comparable for CI-only (7.0±5.9 dB) and bimodal listening (5.9±4.0 dB). In the present study, stimuli were presented from a single loudspeaker (i.e., without spatial cues). As such, any binaural benefits were due to summation effects, which are generally small. It would be worthwhile to study the benefits of monaural or binaural auditory training on binaural perception when spatial cues are available. Data from more participants are needed to better determine if training benefits differ between monaural and binaural training and between monoaural and binaural speech perception.

## Other observations

There was no significant training benefit for SSQ scores. This suggests that the training benefits observed with the lab-based outcome measures did not generalize to the listening scenarios captured by the SSQ questionnaire. Note that the questionnaire reflected hearing abilities over a longer term for a variety of scenarios with participants' everyday listening mode. The speech outcome measures in quiet or with steady noise do not reflect the complex listening environments encountered in everyday life which the SSQ questionnaire seeks to address.

While significant benefits were observed for the present computer-based phonemic contrast training, other approaches have also been shown to be beneficial. Previous studies have shown significant training benefits using a connected discourse training protocol [30, 31]; similar benefits were observed between labor-intensive in-person and computer-based connected discourse training [32]. Oba et al. [11] found significant benefits for digits-in-noise training.

While the present training outcomes are promising, there were only 15 participants in the present study, too few to draw strong conclusions. As such, we regard this as a worthwhile pilot study, with some major limitations. The stimuli and method for measuring SRTs was problematic, as discussed above. Using different outcome measures (e.g., the French Matrix [33] or the FrBio [34]) may help to stabilize pre-training SRTs, avoid repetition of sentences, and better observe training benefits. There was no control group in this study, and procedural learning (or the lack thereof) could only be inferred by the lack of significant difference across test runs. Substantial improvement from the first test run was observed in some participants (see S1 and S2 Figs, and Fig 5). More extensive baseline testing until achieving asymptotic performance would better control for procedural learning. Due to time constraints, this was not possible in the present study.

These preliminary data suggest that computer-based home training may be a useful alternative and/or complement to in-person rehabilitation, which is time-intensive and may not always be possible under certain circumstances (e.g., the COVID-19 pandemic and lockdowns). The computer-based training also offers a consistent training experience, which may not always be the case across CI rehabilitation centers, where approaches and speaking voices may differ. A research version of French AngelSound™ was used in the present study, which restricted access to many other training modules (e.g., familiar word identification, environmental sound identification, speech in noise perception, music perception, etc.). The rich training materials in the full version of the software may further benefit French-speaking CI users, and we plan to further evaluate the benefits of French AngelSound™ in a larger group of CI users. Further studies with a larger number of participants, an explicit control group, and better sentence outcome measures are needed to reveal the benefits of computer-based training for more realistic listening conditions.

## Supporting information

**S1 Fig. Vowel recognition scores for individual CI participants as a function of test run.** Pre-training (left column), post-training (middle column), and follow-up data (right column) are shown for the unilateral group (top row), the bilateral group with the poorer CI ear and binaural listening (middle two rows), and the bimodal group with CI-only and binaural listening (bottom two rows).
(TIF)

**S2 Fig. Consonant recognition scores for individual CI participants as a function of test run.** Pre-training (left column), post-training (middle column), and follow-up data (right column) are shown for the unilateral group (top row), bilateral group with binaural listening (middle row), and the bimodal group with CI-only and binaural listening (bottom two rows).
(TIF)

**S1 File. Dataset comporting all raw data from the present study.** Dataset include tables presenting CI training time, CI vowels and consonants recognition, CI SRTs, CI SSQs and NH data.
(XLSX)

## Acknowledgments

We thank all the participants for their time and effort. We also thank Julie Foucaut, Marion Marguet-Isambert, Antoine Papazian and Dr. Charles Aussedat for their assistance with recording the speech stimuli for French AngelSound™, and Mathieu Robier and Flavie Lemieux for their help with the study.

## Author Contributions

**Conceptualization:** Sandrine Kerneis, John J. Galvin, III, Qian-Jie Fu, David Bakhos.

**Data curation:** Sandrine Kerneis.

**Formal analysis:** John J. Galvin, III.

**Investigation:** Sandrine Kerneis, Jean Baqué.

**Methodology:** Sandrine Kerneis, John J. Galvin, III, Stephanie Borel, Qian-Jie Fu, David Bakhos.

**Project administration:** Sandrine Kerneis, John J. Galvin, III, Qian-Jie Fu, David Bakhos.

**Supervision:** Sandrine Kerneis, John J. Galvin, III, Qian-Jie Fu, David Bakhos.

**Validation:** Sandrine Kerneis, John J. Galvin, III, Stephanie Borel, Qian-Jie Fu.

**Visualization:** John J. Galvin, III.

**Writing – original draft:** Sandrine Kerneis, John J. Galvin, III.

**Writing – review & editing:** Sandrine Kerneis, John J. Galvin, III, Stephanie Borel, Jean Baqué, Qian-Jie Fu, David Bakhos.

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
