## [Decision Letter · Decision Letter 0]

15 Jun 2022

PONE-D-22-10674Benefits of computer-assisted home training for French cochlear implant patientsPLOS ONE

Dear Dr. Kerneis,

Thank you for submitting your manuscript to PLOS ONE. After careful consideration, we feel that it has merit but does not fully meet PLOS ONE’s publication criteria as it currently stands. The reviewers had the impression that the manuscript was improved significantly. Therefore, we invite you to submit a revised version of the manuscript that addresses the points raised during the review process.

We look forward to receiving your revised manuscript.

Kind regards,

Andreas Buechner

Academic Editor

PLOS ONE

Journal Requirements:

Reviewers' comments:

Reviewer's Responses to Questions

**Comments to the Author**

1. Is the manuscript technically sound, and do the data support the conclusions?

Reviewer #1: Partly

Reviewer #2: Yes

2. Has the statistical analysis been performed appropriately and rigorously? 

Reviewer #1: Yes

Reviewer #2: I Don't Know

3. Have the authors made all data underlying the findings in their manuscript fully available?

Reviewer #1: Yes

Reviewer #2: Yes

4. Is the manuscript presented in an intelligible fashion and written in standard English?

Reviewer #1: Yes

Reviewer #2: Yes

5. Review Comments to the Author

Reviewer #1: The resubmitted manuscript is substantially improved in several respects, such as the inclusion of figures showing data from each test run and a greater level of detail on the Methods (though there are still some aspects here that remain a little unclear). However, there are still some major points of concern, most notably regarding the sentence data. Problems here include the fact that listeners would on occasions be presented with sentences that they would have already heard in previous test runs; there is no control for procedural learning, with data from all test runs included despite there being instances in some listeners of steep improvements in performance from the first to the second pre-training test run; missing data due to unmeasurable SRTs suggest that some of the analyses are based on data from only 3 listeners. In my view the sentence data do not seem reliable enough to support any firm conclusions regarding the possible benefit of training. I think it would perhaps be reasonable to include the sentence data unanalysed and to say something along the lines that while they suggest a possible benefit of training, no firm conclusions can be drawn. The overall conclusions would then also need to acknowledge that the study provides no real evidence of generalisation of training benefit to tasks that are anywhere close to real-world listening.

Other issues include:

It seems like overkill to include for the same data a line graph of individual data, bar charts of both mean performance and benefit and also a table.

There are inconsistencies between the data presented in the figures and tables. As one example, Fig 4 shows CI-only vowel recognition in the BM group to be slightly lower at follow-up than at post-training, but benefit is shown as higher at follow-up and Table 3 shows substantially higher performance at follow-up than at post-training.

It is hard to see the relevance of the NH data and I recommend that they be omitted.

Other comments

L185 – Since within-subject data would be collected from both trained and untrained groups, I don’t think it’s appropriate to use across-subject variability as a justification for the absence of a control group. Perhaps more importantly here, the reference to a ‘within-subject control procedure’ presumably refers to the fact that several test runs were conducted prior to training. However, no criteria for assessing whether performance had reached asymptote are mentioned and data from all test runs are included in the pre-training scores, so that no control appears to have been implemented.

L202 – one stimulus

L217 – Would be as well to clarify that this (presumably) means that the recordings were made for this study, rather than that they were new sentences.

L223 – ‘During testing, a sentence was randomly selected from the stimulus set (without replacement)’ – Presumably this refers to each test run, but it would be good to clarify this.

L271 – Would it be more accurate to say ‘4 weeks’, rather than ‘1 month’?

L277 – post-training benefit

L355 – What hypothesis regarding test run was being tested here? Is 5 listeners really sufficient for these analyses?

L477 – The purpose of this analysis isn’t really clear.

L498 - Fig 7 y-axis label should be SRT (dB).

L515 – It appears that, due to the unmeasurable SRTs, the reported mean values include different numbers of listeners at different time points, which is rather misleading.

L516 – ‘For example, when CI-only listening SRTs <25 dB SNR could not be obtained,’ – it appears that the end of this sentences is missing.

L521 – Confusion between lower and higher?

L555 – ‘All participants were tested in the lab.’ This statement doesn’t seem to belong here.

L564 – Why is subsection included as a factor?

L632 – What is the basis for the claim regarding increased performance asymmetry in bilateral users?

L640 – The claim here appears to depend on an assumption that performance would not have improved for the acoustic ear only, but this is just an assumption and not tested.

L652 - The analysis referred to in Table 8 included the effect of test run within each test session, though. With respect to whether procedural learning contributed to improved performance after training the key issue is whether such learning differentially affected performance across different test sessions.

L655 - It appears guaranteed that this was the case, rather than just likely.

L658 – I think it would be beneficial regardless of the details of the adaptive procedure.

L661 – Given the large variability across individual CI users and the different talkers, etc., I don’t think that this comparison across studies is relevant here.

L674 – It’s not clear what lists are being referred to here, if sentences were randomly selected from the full set of 100 on each run.

L689 – Should this say binaural benefit?

L721 – I’m sure it’s true that a lab-based sentence recognition in speech-shaped noise task falls some way short of adequately reflecting realistic listening situations for CI users. However, it does seem hard to believe that training that was truly bringing about a 10 dB improvement in SRT in such a task (e.g., as seen in binaural conditions for BM-5) through perceptual learning rather than through extraneous factors would have no beneficial impact at all on responses to the SSQ.

Reviewer #2: Thanks for this manuscript which is detailed and well written.

I usually prefer person-first language i.e. ‘people with CI’ rather than CI patients …

Lots of journals insist on this now so it’s a good habit to get into

Angel sound been around for years- well proven in other languages – why is doing it in French different?

Children?

Is Angel Sound an App?

Line 121 – is there a reference for this? ‘While 121 unilateral CI patients may adapt to an intra-aural frequency mismatch over time, bimodal CI 122 patients may have difficulty adapting to the mismatch in the CI ear, given the dominance of the 123 acoustic hearing ear’

May be good to do a more formal hypothesis and research question – rather than the less formal ‘what you expect’ - especially as you gave a lot of detailed stats later

Did you invite all eligible adults to participate? Do you think you had a sample representative of the adult CI population?

Did you plan for equal split of unilat/bilat/bimod?

Were patients compensated?

Make sure you write appropriate © or ® when you put the company names

Post pre in your table – post or pre lingual deafness I assume?

Make sure to define all abbrev the first time e.g. SRT in line 175

One ‘stimulus’ not ‘stimuli’ e.g. line 202

Am I right in thinking that if you are presenting stimuli ‘without replacement’, it will become easier to identify the final ones, as there is less choice? Why did you choose this rather than at random?

217 I note that the French HINT sentences were recorded for this study – was validation and equalisation of sentences done to ensure equal difficulty? I know tat can sometime be quite a long process. Why did you choose to rerecord? Was it keyword scoring, or 50% of all words?

Why did you include the listeners with NH? Can you explain please?

657

257 I don’t understand why you say there were lots not used in this study? ‘there are more than 1000 French

257 monosyllable words produced by 2 male and 2 female native French talkers that were not used for

258 testing’

Lines 271 and 330 – repetition of info

I would start results section saying how many people complied, how much training was done e.g. a range of 10 hours to 20 hours with a median of …

How can we show that it’s the training effect not practice at the tests? So does making the ear performance more equal benefit people ultimately or not?

Lots of tables of number may be better in appendices – really loses the readability here

Could the raw data be provided online? Ten plus tables of numbers is just too much!

554 – can you say the range of training done e.g. 9 to 14 hours? Otherwise it seems meaningless to do a correlation between training benefit and amount of training, if they’ve all done pretty much 10 hours.

Can you comment on retention of benefit past one month? Were people keen to continue?

657 50% OF the words

Fig 1 is useful thanks

6. PLOS authors have the option to publish the peer review history of their article (what does this mean?). If published, this will include your full peer review and any attached files.

Reviewer #1: No

Reviewer #2: **Yes: **Helen Cullington

---

## [Author Response · Author response to Decision Letter 0]

18 Aug 2022

To the editor,

We thank the reviewers and the editor for their helpful comments and suggestions and have extensively revised the MS with these comments in mind. Due the extensive revision, changes in the marked-up copy are shown in red type, rather than simply using track changes, which would have made for difficult reading. We can provide the tracked changes version if needed. Major changes include:

1. We have re-evaluated the test run data across groups for the pre-training, post-training, and follow-up measures for CI-only and binaural vowel and consonant recognition. At the suggestion of Reviewer 1, we did not analyze the SRT test run data, due to the limited number of participants for whom SRTs < 25 dB could be obtained. Results showed no significant effect of test run for pre-training vowel and consonant recognition, suggesting stable baseline measures with which to compare post-training improvements within participants. Some effects of test run were observed for post-training and follow-up measures, and data were excluded from the figures and analyses accordingly.

2. As suggested by Reviewer 1, we have removed the mean SRT figure and analyses, and now simply show the test run figure without statistical analysis. Instead, we broadly describe the patterns of results.

3. We have moved the vowel and consonant test run figures and analyses to supplementary material. We also provide all raw data as supplementary data.

4. We have added new Figure 6 to compare monaural and binaural pre-training, post-training, and follow-up vowel and consonant recognition scores for bilateral and bimodal CI users. This new section of the results forms the basis of the discussion regarding whether CI-only training differentially affected CI-only or binaural performance (it did not).

5. We have edited the MS to reduce paper length and improve clarity.

6. We have softened conclusions regarding generalization of the training, especially for sentence recognition in noise.

7. To emphasize the preliminary nature of these results, we have re-titled the paper as “Preliminary evaluation of computer-assisted home training for French cochlear implant recipients.”

We hope you find these changes acceptable. Please contact me if you need further information. Below, we respond to specific reviewer comments.

Sincerely,

Sandrine Kerneis

Reviewer #1

The resubmitted manuscript is substantially improved in several respects, such as the inclusion of figures showing data from each test run and a greater level of detail on the Methods (though there are still some aspects here that remain a little unclear). However, there are still some major points of concern, most notably regarding the sentence data. Problems here include the fact that listeners would on occasions be presented with sentences that they would have already heard in previous test runs; there is no control for procedural learning, with data from all test runs included despite there being instances in some listeners of steep improvements in performance from the first to the second pre-training test run; missing data due to unmeasurable SRTs suggest that some of the analyses are based on data from only 3 listeners. In my view the sentence data do not seem reliable enough to support any firm conclusions regarding the possible benefit of training. I think it would perhaps be reasonable to include the sentence data unanalysed and to say something along the lines that while they suggest a possible benefit of training, no firm conclusions can be drawn. The overall conclusions would then also need to acknowledge that the study provides no real evidence of generalisation of training benefit to tasks that are anywhere close to real-world listening.

>> We have re-analyzed the test run data and found no significant effect of test run across groups for baseline vowel, consonants, and SRTs (after excluding 2 unilateral and 2 bimodal CI users where SRTs <25 dB could not be obtained). This suggests little procedural learning for pre-training measures. For post-train CI-only vowels, significant effects of test run were observed for unilateral and bilateral CI users; for follow-up measures, a significant effect of test run was observed for all groups. For CI-only consonants, there was no significant effect of test run for pre-train, post-train, or follow-up measures. For all binaural measures, there was no significant effect of test run. We have added the results of these analyses in new supplementary files S1 Table.docx and S2 Table.docx. For mean calculations, figures, and subsequent analyses, data were excluded according to the new test run analyses, and are explicitly indicated in the text. Overall, while there was no explicit threshold for asymptotic performance, the data are generally stable, especially for pre-training measures, suggesting that procedural learning effects may have been minimal.

 We have modified the reporting and discussion of the SRT results with your comments in mind. We no longer present statistical analyses and we have removed the figure showing mean performance. We have also modified the end of the MS: “Further studies with a larger number of participants, an explicit control group, and better sentence outcome measures are needed to reveal the benefits of computer-based training for more realistic listening conditions.”

Other issues include:

It seems like overkill to include for the same data a line graph of individual data, bar charts of both mean performance and benefit and also a table.

>>We have moved the individual test run figures to supplementary materials. We now only have Table 1 in the MS, and provide the test run analyses as supplementary material. 

There are inconsistencies between the data presented in the figures and tables. As one example, Fig 4 shows CI-only vowel recognition in the BM group to be slightly lower at follow-up than at post-training, but benefit is shown as higher at follow-up and Table 3 shows substantially higher performance at follow-up than at post-training.

>> We have re-examined and re-analyzed all data and figures using data according to the test run analyses, resulting in some small corrections. 

It is hard to see the relevance of the NH data and I recommend that they be omitted.

>> The NH data in the figures and MS show that the recorded French vowel and consonant stimuli and tests produced near perfect performance, and that the range of SRTs was similar to that with other materials in NH listeners. They also contextualize CI performance for these test measures. Unless there is a strong objection, we prefer to retain the NH data lines in new Figs 3-5 and the short paragraph in the Results.

Other comments

L185 – Since within-subject data would be collected from both trained and untrained groups, I don’t think it’s appropriate to use across-subject variability as a justification for the absence of a control group. Perhaps more importantly here, the reference to a ‘within-subject control procedure’ presumably refers to the fact that several test runs were conducted prior to training. However, no criteria for assessing whether performance had reached asymptote are mentioned and data from all test runs are included in the pre-training scores, so that no control appears to have been implemented.

>>We have re-conducted the test run analyses, now comparing test run and group factors within the pre-train, post-train and follow-up measures for vowels and consonants with CI-only and binaural listening; the complete results of these new analyses are shown in supplementary files S-Table 1.docx and S-Table 2.docx. For pre-training CI-only and binaural vowel recognition, there was no significant effect of test run. For CI-only post-training vowel recognition, performance was significantly poorer for runs 1 and 2 for unilateral CI users, and for run 1 for bimodal CI users. For CI-only follow-up vowel recognition, performance was significantly poorer for run 1 for all groups. Vowel data were excluded from the revised figures and analyses according to the test run analyses. For consonant recognition, there was no significant effect of test run for any group at any of the test points. Given the limited data, test run was not analyzed for SRTs, as suggested by the reviewer. Given these results, we feel that pre-training measures were stable and that the within-subject control approach was valid. We have added to the Discussion: “There was no control group in this study, and procedural learning (or the lack thereof) could only be inferred by the lack of significant difference across test runs. Substantial improvement from the first test run was observed in some participants (see S-Fig 1.pdf, S-Fig 2.pdf, and S-Fig 3.pdf). More extensive baseline testing until achieving asymptotic performance would better control for procedural learning. Due to time constraints, this was not possible in the present study. “

L202 – one stimulus

>> Corrected

L217 – Would be as well to clarify that this (presumably) means that the recordings were made for this study, rather than that they were new sentences.

>> Revised as: “The sentence stimuli used for this study were new recordings produced by the same male and female talkers that produced the vowel and consonant test stimuli (44.1 kHz sampling rate, 16-bit resolution).”

L223 – ‘During testing, a sentence was randomly selected from the stimulus set (without replacement)’ – Presumably this refers to each test run, but it would be good to clarify this.

>> Revised as: “The total stimulus set consisted of 100 sentences. For each test run, 20 sentences were randomly selected from the total set. During testing, a sentence was randomly selected from the 20-sentence set (without replacement) and presented at the designated SNR.”

L271 – Would it be more accurate to say ‘4 weeks’, rather than ‘1 month’?

>> Modified as suggested

L277 – post-training benefit

>> Corrected

L355 – What hypothesis regarding test run was being tested here? Is 5 listeners really sufficient for these analyses?

>> The hypothesis tested was that there would be no differences in scores across test runs. As noted above, we have re-conducted the test run analyses, now comparing test run and group factors within the pre-train, post-train and follow-up measures for vowels and consonants with CI-only and binaural listening; the complete results of these new analyses are shown in supplementary files S-Table 1.docx and S-Table 2.docx.

L477 – The purpose of this analysis isn’t really clear

>> We wanted to see whether the phonemic contrast training differently affected vowel and consonant recognition, and found an overall greater improvement in vowel recognition. Revised as: “Training benefits were also compared between vowel and consonant recognition. LMM analysis was performed on the training benefit data with test (post-train, follow-up), and measure (vowels, consonants) as fixed factors and participant as a random factor. Results showed a significant effect for measure [F(1, 72) = 8.8, p = 0.004], but not for test [F(1, 72) = 1.1, p = 0.289]; there was no significant interaction. Post-hoc Bonferroni pair-wise comparisons showed that the training benefit was significantly larger for vowels than for consonants (p < 0.05).”

L498 - Fig 7 y-axis label should be SRT (dB).

>> Oops. Corrected

L515 – It appears that, due to the unmeasurable SRTs, the reported mean values include different numbers of listeners at different time points, which is rather misleading.

>> Considering your previous comments regarding SRT data, we no longer present mean SRT data or statistical analyses of SRT data. 

L516 – ‘For example, when CI-only listening SRTs <25 dB SNR could not be obtained,’ – it appears that the end of this sentences is missing.

>> We have revised this section and this sentence no longer appears.

L521 – Confusion between lower and higher?

>> Corrected

L555 – ‘All participants were tested in the lab.’ This statement doesn’t seem to belong here.

>> We have deleted this sentence, and we have moved this section to the beginning of the Results, as suggested by Reviewer 2.

L564 – Why is subsection included as a factor?

>> Subsection was included to observe any significant effects of training on the different aspects of the SSQ. We have removed the total SSQ score analysis, as the group and test comparisons were made with the subsection analysis.

L632 – What is the basis for the claim regarding increased performance asymmetry in bilateral users? 

L640 – The claim here appears to depend on an assumption that performance would not have improved for the acoustic ear only, but this is just an assumption and not tested.

>> In response to both comments, we have conducted additional analyses to observe any training effects on binaural advantage (calculated as the difference between bilateral and CI-only performance with the better ear for bilateral CI users, or as the difference between bimodal and CI-only listening for bimodal CI users). We have added a new section and figure to the Results. Analysis showed no significant difference in binaural advantage across pre-training, post-training, and follow-up measures, suggesting that training the CI ear alone similarly improved CI-only and bilateral/bimodal performance. With regards to testing the acoustic ear, we have revised this aspect of the Discussion as: “Training the CI-ear alone produced similar improvements in CI-only and binaural performance (Fig. 6). This suggests that post-training binaural improvements were likely driven by the improvements in the trained ear. Interestingly, there was no significant difference in binaural benefits across the bilateral and bimodal groups, between vowels and consonants, or among the different test points. As such, the performance asymmetry observed in bilateral and bimodal CI users before training largely remained after training despite the improvements in performance. Note that performance was not measured with the acoustic ear alone in bimodal CI users; we assumed that according to CI candidacy requirements, performance in the acoustic ear was sufficiently poor to support cochlear implantation, and remained poorer than that with the CI ear after implantation. It is unclear whether binaural training would have elicited similar or better improvements in CI-only and binaural listening. For example, Chavant et al. [28] found that for simulations of bimodal listening, including low-frequency acoustic information (even though unintelligible) during training benefited perception of spectro-temporally degraded speech.” 

L652 - The analysis referred to in Table 8 included the effect of test run within each test session, though. With respect to whether procedural learning contributed to improved performance after training the key issue is whether such learning differentially affected performance across different test sessions.

>>As noted above, we reconducted the test run analyses for the vowel and consonant data. Consistent with your comments regarding the SRT data, we did not perform any statistical analyses or present mean data. We have also greatly revised the SRT section of the Discussion: “ Pre-training CI-only SRTs < 25 dB could not be obtained in participants UNI-1, UNI-5, BM-1, and BM-2; pre-training binaural SRTs < 25 dB could not be obtained in BM-1. However, SRTs < 25 dB could be obtained in these participants at post-training and follow-up. These improvements in SRTs were sometimes suspiciously large (e.g., >20 dB improvement for UNI-5 at post-training and follow-up), and much greater than would be expected from the phonemic contrast training. In participants for whom CI-only pre-training SRTs <25 dB could be obtained, mean post-training improvements were 5.9±2.1 and 7.0±5.9 dB for unilateral and bimodal CI users, respectively. For these participants, the mean post-training improvements in CI-only vowel scores were 19.4±6.3 and 23.3±3.2 points for unilateral and bimodal CI users, respectively; the mean post-training improvements in CI-only consonant scores were 8.9±4.3 and 13.5±11.3 points for unilateral and bimodal CI users, respectively. It is unclear whether such improvements in phoneme recognition contributed to the somewhat larger improvements in speech understanding in noise.

L655 - It appears guaranteed that this was the case, rather than just likely. 

>> Revised as: “…some sentences were inevitably repeated across test runs and/or test sessions.”

L658 – I think it would be beneficial regardless of the details of the adaptive procedure.

>> Revised as: “…previous exposure to sentences would have been beneficial.”

L661 – Given the large variability across individual CI users and the different talkers, etc., I don’t think that this comparison across studies is relevant here.

>>We have deleted this section

L674 – It’s not clear what lists are being referred to here, if sentences were randomly selected from the full set of 100 on each run.

>> We have deleted this sentence. In the Methods, we clarify: “The total stimulus set consisted of 100 sentences. For each test run, 20 sentences were randomly selected from the total set. During testing, a sentence was randomly selected from the 20-sentence set (without replacement) and presented at the designated SNR.”

L689 – Should this say binaural benefit?

>> Corrected

L721 – I’m sure it’s true that a lab-based sentence recognition in speech-shaped noise task falls some way short of adequately reflecting realistic listening situations for CI users. However, it does seem hard to believe that training that was truly bringing about a 10 dB improvement in SRT in such a task (e.g., as seen in binaural conditions for BM-5) through perceptual learning rather than through extraneous factors would have no beneficial impact at all on responses to the SSQ.

>> We have expanded discussion of the limitations to the study: “While the present training outcomes are promising, there were only 15 participants in the present study, too few to draw strong conclusions. As such, we regard this as a worthwhile pilot study, with some major limitations. The stimuli and method for measuring SRTs was problematic, as discussed above. Using different outcome measures (e.g., the French Matrix [34] or the FrBio [35]) may help to stabilize pre-training SRTs, avoid repetition of sentences, and better observe training benefits. There was no control group in this study, and procedural learning (or the lack thereof) could only be inferred by the lack of significant difference across test runs. Substantial improvement from the first test run was observed in some participants (see S-Fig 1.pdf , S-Fig 2.pdf, and S-Fig 3.pdf). More extensive baseline testing until achieving asymptotic performance would better control for procedural learning. Due to time constraints, this was not possible in the present study.

These preliminary data suggest that computer-based home training may be a useful alternative and/or complement to in-person rehabilitation, which is time-intensive and may not always be possible under certain circumstances (e.g., the COVID-19 pandemic and lockdowns). The computer-based training also offers a consistent training experience, which may not always be the case across CI rehabilitation centers, where approaches and speaking voices may differ. A research version of French AngelSoundTM was used in the present study, which restricted access to many other training modules (e.g., familiar word identification, environmental sound identification, speech in noise perception, music perception, etc.). The rich training materials in the full version of the software may further benefit French-speaking CI users, and we plan to further evaluate the benefits of French AngelSoundTM in a larger group of CI users. Further studies with a larger number of participants, an explicit control group, and better sentence outcome measures are needed to reveal the benefits of computer-based training for more realistic listening conditions.

 

Reviewer #2

Thanks for this manuscript which is detailed and well written.

I usually prefer person-first language i.e. ‘people with CI’ rather than CI patients …Lots of journals insist on this now so it’s a good habit to get into

>> We agree that first-person language is preferable; however, such phrasing would be quite awkward for much of the MS. We have removed all instances of the word “patients.” We now use “CI recipients,” “CI users,” and “participants” where appropriate; these terms appear to be acceptable in more than 150 publications listed in PubMed over the last year. We also changed the terminology to be “adults with NH” or “listeners with NH,” as appropriate.

Angel sound been around for years- well proven in other languages – why is doing it in French different?

>> We wanted to make this rehabilitation software available to French people with CIs, as there currently is no French language computer-based software for home training. We developed and recorded the stimuli, developed the software, and then evaluated French Angel Sound in a group of experienced CI users. Our ultimate goal is to distribute it freely for CI recipients, clinicians, and researchers, as we have with the other versions of Angel Sound.

Children?

>> We did not evaluate French Angel Sound in children in the present study. However, there is a training module for children in the complete version of the software. 

Is Angel Sound an App? 

>> Angel Sound is currently computer-based Windows software but is being developed for use on mobile devices.

Line 121 – is there a reference for this? ‘While unilateral CI patients may adapt to an intra-aural frequency mismatch over time, bimodal CI patients may have difficulty adapting to the mismatch in the CI ear, given the dominance of the acoustic hearing ear’

>> We have added a reference.

May be good to do a more formal hypothesis and research question – rather than the less formal ‘what you expect’ - especially as you gave a lot of detailed stats later

>> We have added: “Some of our research questions were: Would training with a single CI would also improve binaural performance? And, would binaural improvements be super-additive (greater than CI-only improvements) or simply driven by CI-only improvements?”

Did you invite all eligible adults to participate? Do you think you had a sample representative of the adult CI population?

>> Yes, all eligible adults were first seen by the audiologist in the beginning of the enrollment period and were invited to participate in the study. We feel that we had a representative sample, with relatively good and bad performers of different ages, with different amounts of CI experience, and users of different commercial CI devices. 

Did you plan for equal split of unilat/bilat/bimod?

>> Because of the extensive time needed for training, we initially wanted to enroll 15 patients with a goal towards equal numbers of unilateral, bilateral, and bimodal CI users. 

Were patients compensated?

>> No.

Make sure you write appropriate © or ® when you put the company names

>> Corrected. We have also added “TM” for Angel Sound.

Post pre in your table – post or pre lingual deafness I assume?

>> Corrected.

Make sure to define all abbrev the first time e.g. SRT in line 175

>> SRT is now defined at the first instance.

One ‘stimulus’ not ‘stimuli’ e.g. line 202

>> Corrected

Am I right in thinking that if you are presenting stimuli ‘without replacement’, it will become easier to identify the final ones, as there is less choice? Why did you choose this rather than at random? 

>> Presenting stimuli randomly without replacement allows for all stimuli in a stimulus set to be presented an equal number of times; this is standard procedure for closed set tasks. 

217 I note that the French HINT sentences were recorded for this study – was validation and equalisation of sentences done to ensure equal difficulty? I know that can sometime be quite a long process. Why did you choose to rerecord? Was it keyword scoring, or 50% of all words?

>> While the original French HINT sentences and recordings were balanced in loudness to achieve equal intelligibility at a fixed SNR, this was done among adults with NH. Such sentence level adjustments are not relevant for CI listeners (e.g., due to differences in spectral resolution, the effects of the acoustic-to-electric amplitude mapping, etc.). Indeed, only the AZ-Bio sentence lists have been adjusted for CI listeners. For the present study, we simply used the sentence materials from the French HINT. We chose to record new productions because the original French HINT sentences were recorded from French-speaking Canadians, rather than native French speakers. In terms of scoring, it was 50% of all words (we have clarified in the revised MS). 

Why did you include the listeners with NH? Can you explain please?

>> As noted above, we wanted to establish that the recorded French vowel and consonant stimuli produced nearly perfect performance, and that the range of SRTs was similar to that with other materials in listeners with NH. The NH data also contextualize CI performance for these test measures. 

257 I don’t understand why you say there were lots not used in this study? ‘there are more than 1000 French monosyllable words produced by 2 male and 2 female native French talkers that were not used for testing’ 

>> To clarify, revised as: “For the phonemic contrast training used in the present study, there were more than 1000 French monosyllable words produced by 2 male and 2 female native French talkers; these training stimuli were not used for testing.”

Lines 271 and 330 – repetition of info

>> We have consolidated the two paragraphs that describe the training protocol and instructions.

I would start results section saying how many people complied, how much training was done e.g. a range of 10 hours to 20 hours with a median of …

>>We still begin the Results section with short paragraph regarding results from listeners with NH, then follow with a paragraph describing the compliance with the test and training protocols

How can we show that it’s the training effect not practice at the tests? So does making the ear performance more equal benefit people ultimately or not?

>> As noted above, we have re-conducted the test run analyses, and adjusted the data used in the figures and subsequent analyses accordingly. Importantly, there was no significant effect of test run for pre-training vowel and consonant recognition with CI-only or binaural listening. This suggests that procedural learning may have been minimal.

Regarding performance asymmetry, we have revised this section of the Discussion as: “Training the CI-ear alone produced similar improvements in CI-only and binaural performance (Fig. 6). This suggests that post-training binaural improvements were likely driven by the improvements in the trained ear. Interestingly, there was no significant difference in binaural benefits across the bilateral and bimodal groups, between vowels and consonants, or among the different test points. As such, the performance asymmetry observed in bilateral and bimodal CI users before training largely remained after training despite the improvements in performance. Note that performance was not measured with the acoustic ear alone in bimodal CI users; we assumed that according to CI candidacy requirements, performance in the acoustic ear was sufficiently poor to support cochlear implantation, and remained poorer than that with the CI ear after implantation. It is unclear whether binaural training would have elicited similar or better improvements in CI-only and binaural listening. For example, Chavant et al. [28] found that for simulations of bimodal listening, including low-frequency acoustic information (even though unintelligible) during training benefited perception of spectro-temporally degraded speech.”

Lots of tables of number may be better in appendices – really loses the readability here. Could the raw data be provided online? Ten plus tables of numbers is just too much!

>> We have moved the individual test run figures and test run analyses to supplementary materials; we also provide the raw data as supplementary material. We have removed many of the previous tables. Currently, there is one table and 7 figures in the revised MS

554 – can you say the range of training done e.g. 9 to 14 hours? Otherwise it seems meaningless to do a correlation between training benefit and amount of training, if they’ve all done pretty much 10 hours.

>> As noted above, we have moved this paragraph, and we have deleted the correlation reporting.

Can you comment on retention of benefit past one month? Were people keen to continue?

>> Approximately 66% of patients asked if they could keep the software and continue training at home.

657 50% OF the words

>> Corrected

Fig 1 is useful thanks

---

## [Decision Letter · Decision Letter 1]

9 Mar 2023

PONE-D-22-10674R1Preliminary evaluation of computer-assisted home training for French cochlear implant recipientsPLOS ONE

Dear Dr. Kerneis,

Thank you for submitting your manuscript to PLOS ONE. After careful consideration, we feel that it has merit but does not fully meet PLOS ONE’s publication criteria as it currently stands. Therefore, we invite you to submit a revised version of the manuscript that addresses the points raised during the review process.

While one reviewer has accepted the manuscript, there is a strong remaining concern raised by the other reviewer on your method of analyzing the data. The concern is a valid one.

To address the concern, my suggestion would be to adequately discuss the issues of the second reviewer in your manuscript. I see this as a mandatory requirement to get your manuscript published.

We look forward to receiving your revised manuscript.

Kind regards,

Andreas Buechner

Academic Editor

PLOS ONE

Additional Editor Comments (if provided):

Dear Dr. Kerneis,

while one reviewer has accepted the manuscript, there is a strong remaining concern raised by the other reviewer on your method of analyzing the data. The concern is a valid one.

To address the concern, my suggestion would be to adequately discuss the issues of the second reviewer in your manuscript. I see this as a mandatory requirement to get your manuscript published.

Reviewers' comments:

Reviewer's Responses to Questions

**Comments to the Author**

1. If the authors have adequately addressed your comments raised in a previous round of review and you feel that this manuscript is now acceptable for publication, you may indicate that here to bypass the “Comments to the Author” section, enter your conflict of interest statement in the “Confidential to Editor” section, and submit your "Accept" recommendation.

Reviewer #1: (No Response)

Reviewer #2: (No Response)

2. Is the manuscript technically sound, and do the data support the conclusions?

Reviewer #1: Partly

Reviewer #2: Yes

3. Has the statistical analysis been performed appropriately and rigorously? 

Reviewer #1: No

Reviewer #2: Yes

4. Have the authors made all data underlying the findings in their manuscript fully available?

Reviewer #1: Yes

Reviewer #2: Yes

5. Is the manuscript presented in an intelligible fashion and written in standard English?

Reviewer #1: Yes

Reviewer #2: Yes

6. Review Comments to the Author

Reviewer #1: While adequately addressing some key issues the revisions to the ms have introduced new problems. A reasonable case can be made that stable performance over repeated testing pre-training indicates that comparisons between pre- and post-training are unlikely to be much affected by procedural learning. However, the approach of including test run as a factor and then excluding some post-training runs on a post-hoc basis does not seem appropriate to me. Excluding the post-training runs with poorer performance would inevitably seem to result in bias towards finding beneficial effects of training.

The removal of the analysis of the sentence data is not reflected in some parts of the ms. The Abstract mentions sentence in noise recognition as an outcome measure and reports only that performance improved after training, implying that this included sentence recognition. The first sentence of the Discussion states that there were significant benefits for sentence recognition in noise.

Other points

L206 – ‘Data were averaged across all runs’ - This is inconsistent with test run being included as a factor in the analyses.

L238 – The reference to NH testing seems out of place here. I recommended omission of the NH data, but if they are going to be included their purpose needs to be made clear. Also, in the Results it is stated that SRTs were obtained from 13 listeners.

L253 – ‘high-quality recordings…’ – this is stated earlier.

L353 – Make clear whether they were asked or if these were spontaneous requests.

L374 – This suggests that binaural performance was assessed in the unilateral group.

L380 – This should presumably say ‘vowel’ rather than ‘consonant’

L399 and similarly at L441 - It’s not clear exactly which analysis is being referred to here. Were additional analyses performed on differences between pre- and post-training performance?

L428 – ‘all 3 test runs’ – earlier it says there were 2-3 runs

L452 – It’s not clear here exactly which data is included in this analysis but in any case I don’t think that comparing benefit between vowel and consonant tasks is very meaningful or useful. No rationale is provided to expect a difference and given that the tasks differ in various ways such as the numbers of response options and the starting levels of performance I don’t think that finding a significant difference in percentage point benefit would really be informative about changes in underlying perceptual processes.

L473 – SNR rather than SRT

L478 – Again, I’m not convinced that this analysis is really worthwhile.

L513 – Although it’s unlikely to change any of the conclusions, given that it doesn’t appear to be of interest whether scores are higher overall for any particular subsection, it would appear more natural to analyse each subsection separately, rather than to treat subsection as a factor.

L599 – ‘somewhat larger improvements’ – this seems to be comparing the magnitude of an increase in percent correct in a closed-set phoneme recognition task with a change in SRT which doesn’t really seem to make sense.

Reviewer #2: I reviewed this paper originally as reviewer 2.

The work aims to evaluate French Angelsound in 15 people with CI (5 unilateral, 5 bilateral, 5 bimodal). They performed 10 hours training over one month.

Thank you to the authors for considering and mostly including my suggestions in your revised manuscript. This is now a v well written manuscript with a clear abstract; it will contribute to the literature. I love your ‘Other observations’ paragraphs.

There are only 3 things I suggest and they are word changes/copy editing, so I will not need to rereview:

Line 142. I think it’s a bit odd to put ‘some of our research questions were’. Maybe just put that those were your primary research questions.

Line 143 repetition of word ‘would’

line 345 try and ensure there is not a line break as it currently looks like the SRTs were 5.3 on quick glance – can you remove a space perhaps?

7. PLOS authors have the option to publish the peer review history of their article (what does this mean?). If published, this will include your full peer review and any attached files.

Reviewer #1: No

Reviewer #2: **Yes: **Helen Cullington

---

## [Author Response · Author response to Decision Letter 1]

1 Apr 2023

To the editor,

We thank the reviewers and the editor for their helpful comments and suggestions and have revised the MS with these comments in mind. Changes in the marked-up copy are shown using track changes. Major changes include:

• Revised analysis of the vowel and consonant data to include all test runs, as suggested by Rev. 1. The attendant figures and text have been modified accordingly.

• Revised figure and analysis of the binaural advantage data. 

• Revised SSQ analysis as suggested by Rev. 1

We feel that the changes make the paper more straightforward and we hope you find these changes acceptable. Please contact me if you need further information. Below, we respond to specific reviewer comments.

Sincerely,

Sandrine Kerneis

Reviewer #1

While adequately addressing some key issues the revisions to the ms have introduced new problems. A reasonable case can be made that stable performance over repeated testing pre-training indicates that comparisons between pre- and post-training are unlikely to be much affected by procedural learning. However, the approach of including test run as a factor and then excluding some post-training runs on a post-hoc basis does not seem appropriate to me. Excluding the post-training runs with poorer performance would inevitably seem to result in bias towards finding beneficial effects of training.

>> We have redone all vowel and consonant data analysis and figures with all data included. As the story doesn’t change, this is a much cleaner presentation. We thank you for the suggestion.

The removal of the analysis of the sentence data is not reflected in some parts of the ms. The Abstract mentions sentence in noise recognition as an outcome measure and reports only that performance improved after training, implying that this included sentence recognition. The first sentence of the Discussion states that there were significant benefits for sentence recognition in noise.

>> We now specify in the Abstract: “For all participants, post-training CI-only vowel and consonant recognition scores significantly improved after phoneme training with the CI ear alone. For bilateral and bimodal CI users, binaural vowel and consonant recognition scores also significantly improved after training with a single CI ear.”

And later in the Discussion: “The computer-based home training showed significant benefits for vowel and consonant recognition for the present French CI participants, consistent with previous home-training studies with English-, German- and Mandarin-speaking CI users [5,6,7,8,9,10,11,12,14,15].”

Other points

L206 – ‘Data were averaged across all runs’ - This is inconsistent with test run being included as a factor in the analyses.

>> As noted above, we now include all data in the analyses and figures, so this statement is now correct. 

L238 – The reference to NH testing seems out of place here. I recommended omission of the NH data, but if they are going to be included their purpose needs to be made clear. Also, in the Results it is stated that SRTs were obtained from 13 listeners.

>> We prefer to retain the NH data, as it provides a point of comparison for the CI data in Figs 3-5. We have deleted the sentence, but have added to the Participants section: “Sixteen adults with normal hearing (NH; mean age at testing = 42.5±17.9 years) were also recruited as experimental controls for the CI test data; vowel and consonant recognition were measured in 16 listeners with NH, and sentence recognition in noise was measured in 13 listeners with NH.” 

And later in the Results” “However, CI participants’ SRTs remained much poorer than mean SRTs for listeners with NH (-5.3 ± 2.2 dB), even after training.”

L253 – ‘high-quality recordings…’ – this is stated earlier.

>> The sentence has been deleted.

L353 – Make clear whether they were asked or if these were spontaneous requests.

>> Revised as: “Nonetheless, 66% of participants spontaneously asked if they could keep the training software after the experiment was finished; these requests were honored.” 

L374 – This suggests that binaural performance was assessed in the unilateral group.

>> Revised as: “LMM analysis was also performed on the binaural vowel score data, with group (bilateral, bimodal), test point (pre, post, follow-up), and test run (1, 2, 3, 4) as fixed factors and participant as a random factor.”

L380 – This should presumably say ‘vowel’ rather than ‘consonant’

>> Thanks. This section has been greatly modified, and we corrected the mistake.

L399 and similarly at L441 - It’s not clear exactly which analysis is being referred to here. Were additional analyses performed on differences between pre- and post-training performance?

>> For vowels, revised as: “LMM analyses were performed on the CI-only vowel training benefit data, with group (unilateral, bilateral, bimodal), test point (post, follow-up), and test run (1, 2, 3) as fixed factors and participant as a random factor. Results showed no significant main effects. LMM analyses were also performed on the binaural vowel training benefit data, with group (bilateral, bimodal), test point (post, follow-up), and test run (1, 2, 3) as fixed factors and participant as a random factor. Results showed no significant main effects.”

For consonants, revised as: “LMM analysis were performed on the CI-only consonant training benefit data, with group (unilateral, bilateral, bimodal), test point (post, follow-up), and test run (1, 2, 3) as fixed factors and participant as a random factor; results showed no significant main effects. LMM analysis were also performed on the binaural consonant training benefit data, with group (bilateral, bimodal), test point (post, follow-up), and test run (1, 2, 3) as fixed factors and participant as a random factor; results showed no significant main effects.”

L428 – ‘all 3 test runs’ – earlier it says there were 2-3 runs

>> This section has been greatly modified, and we corrected the mistake.

L452 – It’s not clear here exactly which data is included in this analysis but in any case I don’t think that comparing benefit between vowel and consonant tasks is very meaningful or useful. No rationale is provided to expect a difference and given that the tasks differ in various ways such as the numbers of response options and the starting levels of performance I don’t think that finding a significant difference in percentage point benefit would really be informative about changes in underlying perceptual processes.

>> We have deleted this section.

L473 – SNR rather than SRT

>> This has been corrected.

L478 – Again, I’m not convinced that this analysis is really worthwhile.

>> We prefer to retain this section, as we state at the end of the Introduction: “We were interested in whether training a single CI would also improve binaural performance in bilateral and bimodal CI users and, if so, whether binaural improvements would be super-additive (greater than CI-only improvements) or simply driven by CI-only improvements.” We have modified Fig 6 to explicitly show the binaural advantage at each test point, and we have modified the LMM to compare binaural advantage across groups, tests, and test points. 

L513 – Although it’s unlikely to change any of the conclusions, given that it doesn’t appear to be of interest whether scores are higher overall for any particular subsection, it would appear more natural to analyse each subsection separately, rather than to treat subsection as a factor.

>> We have performed the suggested analysis and have modified this section accordingly.

L599 – ‘somewhat larger improvements’ – this seems to be comparing the magnitude of an increase in percent correct in a closed-set phoneme recognition task with a change in SRT which doesn’t really seem to make sense.

>> We have deleted this section of the paragraph.

 

Reviewer #2

I reviewed this paper originally as reviewer 2.

The work aims to evaluate French Angelsound in 15 people with CI (5 unilateral, 5 bilateral, 5 bimodal). They performed 10 hours training over one month.

Thank you to the authors for considering and mostly including my suggestions in your revised manuscript. This is now a v well written manuscript with a clear abstract; it will contribute to the literature. I love your ‘Other observations’ paragraphs.

There are only 3 things I suggest and they are word changes/copy editing, so I will not need to rereview:

Line 142. I think it’s a bit odd to put ‘some of our research questions were’. Maybe just put that those were your primary research questions.

Line 143 repetition of word ‘would’

>> In response to both comments. Revised as: “We were interested in whether training a single CI would also improve binaural performance in bilateral and bimodal CI users and, if so, whether binaural improvements would be super-additive (greater than CI-only improvements) or simply driven by CI-only improvements”

line 345 try and ensure there is not a line break as it currently looks like the SRTs were 5.3 on quick glance – can you remove a space perhaps?

>> This has been corrected.

---

## [Editor Report · Decision Letter 2]

18 Apr 2023

Preliminary evaluation of computer-assisted home training for French cochlear implant recipients

PONE-D-22-10674R2

Dear Dr. Kerneis,

We’re pleased to inform you that your manuscript has been judged scientifically suitable for publication and will be formally accepted for publication once it meets all outstanding technical requirements.

Kind regards,

Andreas Buechner

Academic Editor

PLOS ONE
---

## [Editor Report · Acceptance letter]

20 Apr 2023

PONE-D-22-10674R2 

Preliminary evaluation of computer-assisted
home training for French cochlear implant recipients 

Dear Dr. Kerneis:

I'm pleased to inform you that your manuscript has been deemed suitable for publication in PLOS ONE. Congratulations! Your manuscript is now with our production department. 

Kind regards, 

on behalf of

Andreas Buechner 

Academic Editor

PLOS ONE